# Pleiotropic actions of phenothiazine drugs are detrimental to Gram-negative bacterial persister cells

Sayed Golam Mohiuddin[1], Thao Vy Nguyen[1] & Mehmet A. Orman [ID] [1✉]

Bacterial persister cells are temporarily tolerant to bactericidal antibiotics but are not necessarily dormant and may exhibit physiological activities leading to cell damage. Based on the link between fluoroquinolone-mediated SOS responses and persister cell recovery, we screened chemicals that target fluoroquinolone persisters. Metabolic inhibitors (e.g., phenothiazines) combined with ofloxacin (OFX) perturbed persister levels in metabolically active cell populations. When metabolically stimulated, intrinsically tolerant stationary phase cells also became OFX-sensitive in the presence of phenothiazines. The effects of phenothiazines on cell metabolism and physiology are highly pleiotropic: at sublethal concentrations, phenothiazines reduce cellular metabolic, transcriptional, and translational activities; impair cell repair and recovery mechanisms; transiently perturb membrane integrity; and disrupt proton motive force by dissipating the proton concentration gradient across the cell membrane. Screening a subset of mutant strains lacking membrane-bound proteins revealed the pleiotropic effects of phenothiazines potentially rely on their ability to inhibit a wide range of critical metabolic proteins. Altogether, our study further highlights the complex roles of metabolism in persister cell formation, survival and recovery, and suggests metabolic inhibitors such as phenothiazines can be selectively detrimental to persister cells.

[1] Department of Chemical and Biomolecular Engineering, University of Houston, Houston, TX, USA. ✉email: morman@central.uh.edu

Persister cells are rare phenotypic variants present in various bacteria within a susceptible isogenic population that tolerate lethal concentrations of antibiotics[1,2]. Following antibiotic removal, persister cells can form a subpopulation indistinguishable from the original antibiotic-sensitive population[2]. Bacterial persisters are not drug-resistant mutants, which may divide in the presence of the antibiotic and give rise to subpopulations of drug-resistant progenies[3]. Instead, persister cells may enter a non-proliferative state, before or during antibiotic treatment, survive lethal concentrations of antibiotics, and may reawaken after the antibiotic is removed[4]. Persisters are a substantial healthcare problem, as their high level of antibiotic tolerance represents a reservoir for the subsequent emergence of drug-resistant mutants[5–7] and biofilm-associated recurrent infections[8]. Annually, biofilm-associated infections afflict approximately 17 million people, resulting in approximately 550,000 deaths in the United States alone[9,10].

The mechanisms underlying persister cell formation, survival, and recovery are complex and diverse. Stochastic and deterministic mechanisms associated with various cellular processes can arrest cell growth, including the SOS response[11–13], toxin/antitoxin modules[14,15], and the stringent response[16,17], which may lead to persistence. However, dormancy is not the only survival mechanism used by persister cells[18,19]. In fact, persister cells may use enhanced cellular repair[12] and antibiotic efflux to survive[20]. Moreover, the cellular target of the antibiotic may be inactive in persister cells during active proliferation[18]. Although arrest of cell growth and dormancy are the most common mechanisms underlying persister cell formation, a large body of evidence indicates persisters can be heterogeneous, demonstrating diverse physiological activities. Persister cells may have active electron transport chains[21,22], produce ATP[23,24], and futilely synthesize and degrade RNAs[25]. Even in apparent dormancy, persister cells must sustain a minimum adenylate energy charge[26] and must synthesize and repair cellular components that are denatured or degraded during antibiotic treatment[12,26]. To determine whether the extent of antibiotic-induced damage determines persistence to fluoroquinolones such as ofloxacin (OFX), two groups analyzed the SOS response in single cells of OFX-treated *Escherichia coli* (*E. coli*)[12,13]. They demonstrated OFX treatment induces the same level of damage in persister cells as in non-persister cells that succumbed to the treatment[12,13]. Therefore, persisters must repair the DNA damage for survival[12,13], though the repair machinery is not necessary until the recovery period following removal of the antibiotic[12]. The implications of these studies are: (i) cellular events following antibiotic treatment (persister recovery) are as important for survival as those occurring before treatment (persister formation)[12] and (ii) persisters are capable of responding to external factors, indicating the presence of active cellular processes that may be essential for their recovery[12,13].

The ability of each persister cell to produce a viable colony is the standard for determining persister status. While some metabolic inhibitors may increase antibiotic tolerance by depleting energy molecules and inducing cell dormancy[27,28], other metabolic inhibitors may reduce numbers of persisters by irreversibly damaging their survival and recovery mechanisms. Also, some metabolic inhibitors might be detrimental to persister cells that are already damaged by antibiotics, as persister cells with antibiotic-induced damage are indistinguishable from antibiotic-sensitive subpopulations of cells[12,13]. In this study, we examined *E. coli* persistence to OFX, a fluoroquinolone antibiotic, which damages DNA by binding to DNA gyrase[29]. Based on the link between the SOS response and OFX persistence[12], we developed a rapid, efficient, high-throughput chemical screening method and demonstrated phenothiazine drugs can drastically reduce persister cell levels in OFX-treated cultures. Our in-depth analysis has revealed the effects of phenothiazine drugs on bacteria are highly pleiotropic. They can reduce cellular redox activities and ATP levels, transiently impair cell membrane, and indirectly perturb the expression of cellular repair mechanisms. Additionally, phenothiazine drugs are highly effective against a diverse range of persister populations obtained from various bacterial species or different antibiotic treatments. Collectively, our data support the notion that persister cells sustain a steady state adenylate energy charge to maintain cellular processes essential for their survival (e.g., transcription/translation, DNA repair mechanisms)[26], and perturbation of metabolic mechanisms associated with this model reduces persister cell levels.

## Results

**Designing a rapid and efficient chemical screen.** The fluoroquinolone OFX forms a complex with DNA gyrase that leads to double-strand DNA breaks, which activates the SOS response and leads to the arrest of cell division and induction of DNA repair pathways[30]. We used *E. coli* strains harboring low-copy-number plasmids in which the gene for green fluorescent protein (*gfp*) was fused to the promoters of the SOS genes $P_{recA}$, $P_{recN}$, $P_{sulA}$, and $P_{tisB}$[12]. The cells were grown in Luria-Bertani (LB) medium to early stationary phase ($t = 5$ h) (Supplementary Fig. 1), treated with OFX, and assessed for GFP. After the addition of OFX, the $P_{recA}$-*gfp* reporter strain had the highest levels of GFP expression (Fig. 1a). Consistent with previous studies[12,31], the strain with *recA* deletion had significantly fewer persisters compared to wild type (WT) controls (Fig. 1b).

Given that persister cells respond to antibiotic-induced damage by inducing repair mechanisms essential for their survival[12,13], chemical inhibitors of bacterial repair mechanisms may eliminate persister cells. Several RecA inhibitors have shown promise for the reduction of antibiotic tolerant cells[32], although these inhibitors have not, to our knowledge, been approved by the US Food and Drug Administration (FDA). When we tested a RecA inhibitor, copper phthalocyanine-3,4′,4″,4‴-tetrasulfonic acid tetrasodium salt (Cu-PcTs), we could not eliminate OFX persisters at concentrations from 0.0156–8.0 mM (Supplementary Fig. 2). Although we treated early stationary phase cells with OFX and Cu-PcTs for 20 h, the RecA inhibitor seems to be effective only after prolonged treatments (2–3 days)[32]. Our aim was to identify medicinally relevant drugs that can robustly eliminate the OFX persisters; therefore, we created a rapid and efficient chemical screen based on the link between the SOS response and persistence (Fig. 1a). The high-throughput chemical screening was performed using Phenotype Microarrays (Biolog, Inc.), which contain ~360 FDA-approved drugs, antibiotics, or small molecules in 96-well plates. Cells with the $P_{recA}$-*gfp* reporter grown in LB in flasks were treated with OFX in early stationary phase ($t = 5$ h) (Supplementary Fig. 1), immediately transferred to 96-wells plates containing the various test chemicals, and GFP was measured 4 h later. Cells treated only with OFX and untreated cells were used as positive and negative controls, respectively. Chemicals that inhibited the repair pathways by preventing the induction of *recA* were expected to have low GFP levels, similar to the negative controls. Although several chemicals reduced *recA* expression, we devised three criteria to identify compounds that specifically target persistence mechanisms: (i) inhibition of *recA*, as measured by GFP expression levels in the reporter strain treated with OFX, (ii) significant reduction in persister cell levels, and (iii) low toxicity to cells. Cells from the wells with the chemicals that reduced *recA* expression were collected after the 20 h treatment to determine the number of persister cells. The experiment was repeated in the absence of OFX to identify chemicals that did not themselves reduce cell viability. The chemical library had four concentrations for each

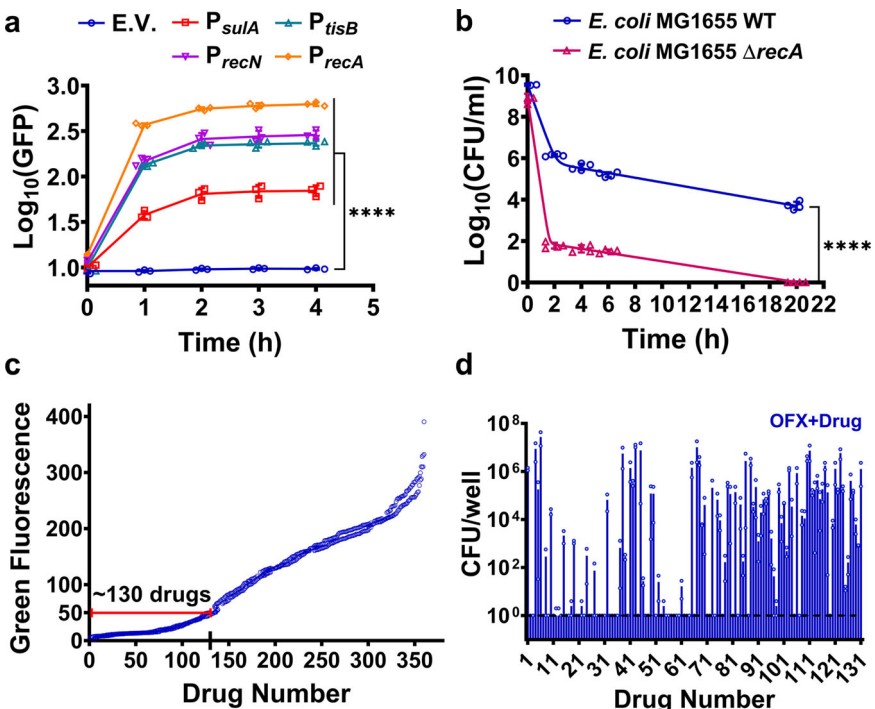

**Fig. 1 Screening assay to identify chemicals that eliminate bacterial persisters. a** Induction of the SOS response by OFX. Cells (*E. coli* MG1655) with reporter plasmids expressing SOS genes were grown in Luria-Bertani (LB) medium, treated with OFX (5 µg/ml) at early stationary phase ($t = 5$ h) and assessed for GFP expression with a plate reader. Control: Empty vector (E.V.). Number of independent biological replicates, $n = 3$. **b** Persister levels in a *recA* mutant. WT or Δ*recA E. coli* MG1655 strains at early stationary phase in LB were treated with OFX for 20 h. Colony forming units (CFU) were determined by plating on LB agar medium. A non-linear model was used to generate the biphasic kill curves. F-statistics were used to perform statistical significance tests. $n = 4$. **c** Drug screening. Early stationary phase cells (*E. coli* MG1655 strain with pUA66 P*recA*-*gfp*) in LB were treated with OFX and immediately transferred to PM plates. GFP measurements were performed after 4 h of incubation. $n = 2$. **d** Persister levels in *E. coli* MG1655 cultures treated with OFX + drug in Phenotype Microarrays. Cells were treated with chemicals in the presence of OFX in PM plates for 20 h and the cultures from the first 130 wells (with the lowest GFP levels) were plated for viable cell counts. Dashed lines represent the limit of detection for CFU counts. $n = 2$. Statistical analysis was performed using one-way ANOVA with Dunnett's post-test (unless otherwise stated), where ****$P < 0.0001$. Data corresponding to each time point represent mean value ± standard deviation.

drug, but these concentrations were not disclosed by the company. As the primary screening involved high cell densities (early stationary phase cultures), we focused on the wells with the highest chemical concentrations. From these experiments, we identified approximately 130 chemicals that inhibited *recA* expression, measured by GFP expression (Fig. 1c, d). However, among these chemicals, only seven fulfilled all three criteria: thioridazine, trifluoperazine, chlorpromazine, amitriptyline, hexachlorophene, pentachlorophenol, and potassium tellurite (Supplementary Figs. 3 and 4).

**Phenothiazine drugs can drastically reduce persister levels at sublethal concentrations.** The seven chemical candidates were further tested for reproducibility, nearly complete inhibition of GFP expression, and reduction of persister cell levels. Early stationary phase cells ($t = 5$ h), grown in LB in flasks, were transferred to test tubes and treated with the candidate chemicals at various concentrations, with or without OFX. Cultures were treated with the candidate chemicals in the absence of OFX to determine whether the chemical itself had antimicrobial activity. Although the tested chemicals inhibited *recA* expression across a wide range of concentrations when tested with OFX (Supplementary Fig. 3), and high concentrations killed the bacteria (Supplementary Fig. 4), lower concentrations (i.e., sublethal concentrations) of the chemicals reduced *recA* expression and eliminated the OFX persisters without affecting cell viability (Supplementary Fig. 4). At these sublethal concentrations, most

of the chemicals reduced persister levels to below the limit of detection (Fig. 2a, b). The chemicals hexachlorophene, pentachlorophenol, and potassium tellurite (Fig. 2a) are antiseptics (e.g., germicide, algaecide, or wood preservative)[33–35], whereas thioridazine (TDZ), trifluoperazine (TFP), chlorpromazine (CPZ), and amitriptyline (Fig. 2a, b) are antipsychotic drugs used to treat depression, schizophrenia, anxiety, and bipolar disorder[36]. TFP, TDZ, and CPZ are phenothiazine drugs, which are safe[37], widely used[36], and administered orally[38]. Further, we tested two additional FDA-approved phenothiazine drugs, fluphenazine (FPZ) and perphenazine (PPZ), that were not included in our drug screening. Both FPZ and PPZ inhibited *recA* expression (Supplementary Fig. 3) and eliminated OFX persisters (Fig. 2c) at concentrations that did not affect cell viability (Supplementary Fig. 4). Moreover, treatment with phenothiazine and norfloxacin, moxifloxacin, levofloxacin, and ciprofloxacin eliminated quinolone persister phenotypes in early stationary phase *E. coli* cultures (Fig. 2d). Phenothiazine also reduced OFX persister levels of *Pseudomonas aeruginosa* strain PA01 (a common opportunistic pathogen that causes pneumonia in cystic fibrosis patients) and highly pathogenic isolates *Klebsiella pneumoniae* (CXY 130) and *Acinetobacter baumannii* (BAA-1605) (Fig. 3). In addition, phenothiazines eliminated ampicillin persisters in early stationary phase *E. coli* cultures (Supplementary Fig. 5), demonstrating their general effectiveness. We previously showed that pretreatment of stationary phase *E. coli* cells with TDZ, TFP, or CPZ significantly reduced type I persisters[39], which are typically formed by their slow exit from the stationary phase in

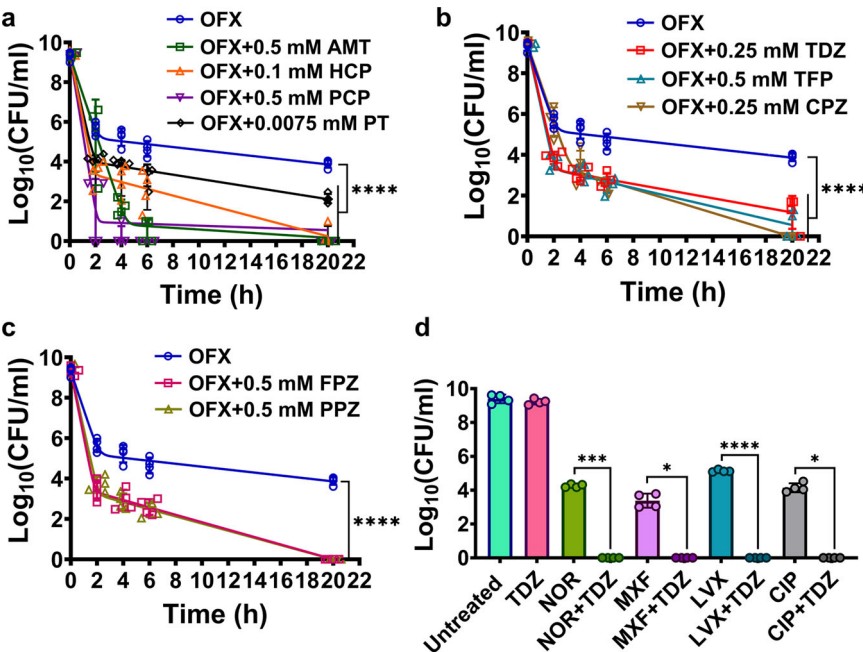

**Fig. 2 Identified chemical hits reduced persister levels in *E. coli* MG1655 cultures. a–c** Biphasic kill curves. Early stationary phase *E. coli* MG1655 cells, grown in LB, were treated with OFX (5 µg/ml) and the chemicals at indicated concentrations. Throughout treatment cells were plated for viable cell counts at indicated time points. Of note, the control data (OFX only) is the same in all three panels. TDZ Thioridazine; TFP Trifluoperazine; FPZ Fluphenazine; PPZ Perphenazine; CPZ Chlorpromazine; AMT Amitriptyline; HCP Hexachlorophene; PCP Pentachlorophenol; PT Potassium tellurite. $n = 4$. **d** Cell survival after quinolone and phenothiazine treatment. Cells (*E. coli* MG1655) treated with indicated quinolone and/or phenothiazines for 20 h were plated for viable cell counts. NOR: norfloxacin (0.8 µg/ml); MXF Moxifloxacin (0.8 µg/ml), LVX Levofloxacin (0.3 µg/ml); CIP Ciprofloxacin (0.2 µg/ml). $n = 4$. Statistical analysis was performed using one-way ANOVA with Dunnett's post-test, where $*P < 0.05$, $***P < 0.001$, $****P < 0.0001$. Data corresponding to each time point represent mean value ± standard deviation.

fresh media[4]. Although we did not demonstrate the underlying mechanism, we speculated that phenothiazine drugs reduced type I persister formation by inhibiting stationary phase metabolism[39]. Phenothiazine drugs are known to interact with a diverse range of membrane-bound proteins[40–44]; however, the specific mechanisms linking these drugs to bacterial persistence remain unknown. Because of their potential therapeutic value, we focused on phenothiazine drugs for the rest of the study to gain a better understanding of their anti-persister characteristics.

**Phenothiazine drugs can reduce cellular transcription/translation activities.** *RecA* expression during recovery, following removal of antibiotics, is essential for persister survival[12,31]. After treating early stationary phase cells (harboring the GFP reporter expressed from the *recA* promoter) with OFX and a phenothiazine drug, we transferred the cells to fresh liquid medium (i.e., recovery culture) and monitored *recA* expression in the cells by fluorescent microscopy (Fig. 4a, b). Cell proliferation in the recovery culture was determined with colony forming unit (CFU) measurements (Supplementary Fig. 6). Our data further showed OFX did not induce *recA* expression in the presence of TDZ and TDZ + OFX-treated cells did not proliferate in fresh medium after the removal of TDZ and OFX (Fig. 4b and Supplementary Fig. 6). In contrast, cells treated only with OFX initiated growth around 6–7 h after their transfer to fresh medium (Supplementary Fig. 6) and had high levels of GFP initially (Fig. 4a). However, some cells formed filaments and continued to express *recA* as measured by GFP (green arrows) during recovery, which is typical of the DNA damage response[12,13]. When the recovery culture entered the mid-exponential phase, GFP was not detected in most dividing cells (blue arrows) (Fig. 4a). On the other hand, GFP positive cells or heterogeneous recovery cell subpopulations were not detected in TDZ + OFX-treated cultures (Fig. 4b).

In addition, we determined the effect of phenothiazine drugs on the expression of other SOS regulon genes, including *sulA*, *tisB*, and *recN*, using the reporter plasmids. In early stationary phase cells, phenothiazine drugs significantly inhibited OFX-induced expression of *sulA*, *tisB*, and *recN* compared to controls treated only with OFX (Supplementary Fig. 7a–c). We also measured synthesis of GFP from an IPTG-inducible synthetic *T5* promoter in pUA66 (a low-copy plasmid) and in pQE-80L (a high-copy plasmid). Phenothiazine drugs significantly inhibited GFP synthesis from these two expression systems compared to the controls (Supplementary Fig. 7d, e), verifying that these drugs impair the general transcription/translation activities, not specifically SOS response genes' expression, in cells.

**Phenothiazine drugs can repress cellular energy metabolism.** Phenothiazine drugs may repress cellular energy metabolism, which may explain the reduced transcription/translation activities in phenothiazine-treated cells (Fig. 4 and Supplementary Fig. 7), as these are energy consuming processes. As cellular energy metabolism seems to be a conserved survival mechanism of persister cells, phenothiazine-mediated repression of energy metabolism may also explain the observed broader impact of these drugs on various persister populations (e.g., ampicillin persisters)[26]. To study the impact of phenothiazines on cellular energy metabolism, untargeted mass spectrometry (MS) (Metabolon, Inc., Durham, NC), which can measure small amounts of a broad spectrum of molecules, was used to quantify the metabolites in phenothiazine-treated cultures compared to controls. *E. coli* MG1655 cells in early stationary phase were treated with TDZ, in the absence of OFX, and were collected 20 h later for MS analysis, along with untreated control cells. Based on the similarity of phenothiazines and their likely similar mechanism of action against bacterial persisters, only TDZ-treated cells were

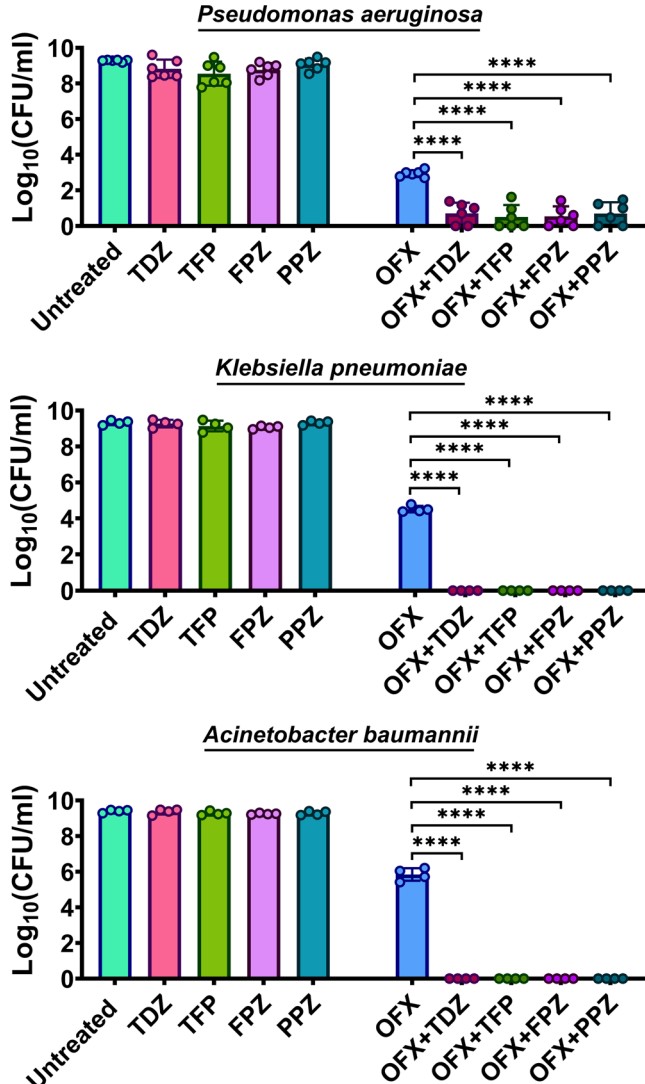

**Fig. 3 Phenothiazine drugs reduced persister levels of Gram-negative bacteria.** Early stationary phase cells of *P. aeruginosa, K. pneumonia,* and *A. baumannii* strains (see Supplementary Fig. 1 for growth curves) were treated with OFX (5 μg/ml, 25 μg/ml, and 140 μg/ml, respectively) and the phenothiazine drugs for 20 h. Then, cultures were plated for viable cell counts. Phenothiazine concentrations for *P. aeruginosa* and *K. pneumoniae*: TDZ: 0.25 mM; TFP: 0.5 mM; FPZ: 0.5 mM; PPZ: 0.5 mM. Phenothiazine concentrations for *A. baumannii*: TDZ: 0.1 mM; TFP: 0.1 mM; FPZ: 0.1 mM; PPZ: 0.1 mM. Of note, *A. baumannii* species was more sensitive to phenothiazines compared to other species. $n \geq 4$. Statistical analysis was performed using one-way ANOVA with Dunnett's post-test, where ****$P < 0.0001$. Data corresponding to each time point represent mean value ± standard deviation.

analyzed by MS. Metabolites from extracts of the TDZ-treated and control cells were identified by ultra-high-performance liquid chromatography-tandem MS, with internal standards, by comparison with an extensive metabolite library. The 496 metabolites that were identified are involved in 8 super-pathways and 76 sub-pathways, including those for amino acids, peptides, carbohydrates, lipids, nucleotides, vitamins, and xenobiotics. Unsupervised hierarchical clustering of the metabolic data matrix indicated that metabolites identified in four independent biological replicates of untreated or TDZ-treated samples clustered together (Fig. 5a), verifying the reproducibility of the data. Metabolites associated with the tricarboxylic acid (TCA) cycle

were significantly downregulated in TDZ-treated cultures compared to controls (Fig. 5b). TDZ treatment also resulted in an accumulation of metabolites associated with glycolysis and carbon metabolism (e.g., glucose, fructose, pyruvate, and maltose), indicating TDZ-treated cells might not efficiently metabolize these compounds to generate energy (Fig. 5c, d). Inhibition of the TCA cycle, the major energy-yielding metabolic pathway[45], by TDZ was confirmed using redox sensor green (RSG) dye (Fig. 5e). RSG emits green fluorescence when it is reduced by bacterial reductases, which are an essential component of energy metabolism (Supplementary Fig. 8a and Supplementary Fig. 9). As expected, TDZ and other phenothiazine drugs significantly decreased green fluorescence emission by RSG in cells, indicating a decrease in cellular energy metabolism (Fig. 5e). Metabolic inhibition in phenothiazine-treated cells should be associated with a reduction in ATP[40,46,47]. Therefore, we measured ATP in early stationary phase cells after phenothiazine treatment and found that untreated cells had significantly more ATP than treated cells (Fig. 5f), confirming phenothiazines are strong metabolic inhibitors.

**Phenothiazine drugs potentially inhibit membrane-bound metabolic proteins.** Phenothiazines reduce intracellular ATP production, which may be due to their interactions with electron transport chain (ETC) proteins[40]. These interactions may also impair membrane integrity[43,48]. To characterize the permeability of cellular membranes, early stationary phase cells were treated with phenothiazine drugs (without OFX) for 20 h; then, samples were collected at indicated time points for propidium iodide (PI) staining, a fluorescent dye that is permeant to cells with compromised membranes. Our data reveal that sublethal concentrations of phenothiazines permeabilized the cellular membranes of the bulk cell populations 1 h after the treatment. However, the impact of the phenothiazine treatments was transient, as the PI positive cell levels were decreased with respect to treatment time (Fig. 6a). Despite this transient permeabilization state, optical density measurements of cultures indicated the cells can continue to grow in the presence of phenothiazines, albeit at slower rates compared to untreated controls (Supplementary Fig. 10). To examine if increased cellular membrane permeability is due to interaction between phenothiazines and membrane-bound proteins, we screened a subset of single gene deletions from *E. coli* (K-12 BW25113) Keio Knockout Collection that lack genes associated with ETC, ATP synthase, efflux pumps, and transporters (see Methods). In our screening assay, the mutant strains were individually grown in test tubes, treated with 0.25 mM TDZ at early stationary phase [as this condition drastically permeabilized the WT *E. coli* strain (Fig. 6a)], then stained with PI 1 h after TDZ treatment for flow cytometry analysis. Our rationale is if there is an interaction between a membrane protein and TDZ, which enhances cellular membrane permeability, membrane permeability should be significantly reduced in the absence of this membrane protein. Although we did not identify a single gene deletion that eliminated the effect of TDZ, the mutant strains that were less permeabilized by TDZ generally lacked genes encoding protein subunits of succinate: quinone oxidoreductase (SdhA, SdhB, SdhC, and SdhD), cytochrome bd-I ubiquinol oxidase (CydX), and NADH:quinone oxidoreductase complexes (NuoJ and NuoF) (Fig. 6b), implying the potential ability of phenothiazine to inhibit a wide range of critical proteins of energy metabolism. When we knocked out some of the identified genes (ΔsdhA, ΔsdhB, ΔsdhC, ΔsdhD, ΔcydX, ΔnuoJ, and ΔnuoF) in the *E. coli* MG1655 strain, we found deletions of these genes generally reduced TDZ-induced membrane permeabilization in *E. coli* MG1655 mutant strains at early stationary phase (Fig. 6c),

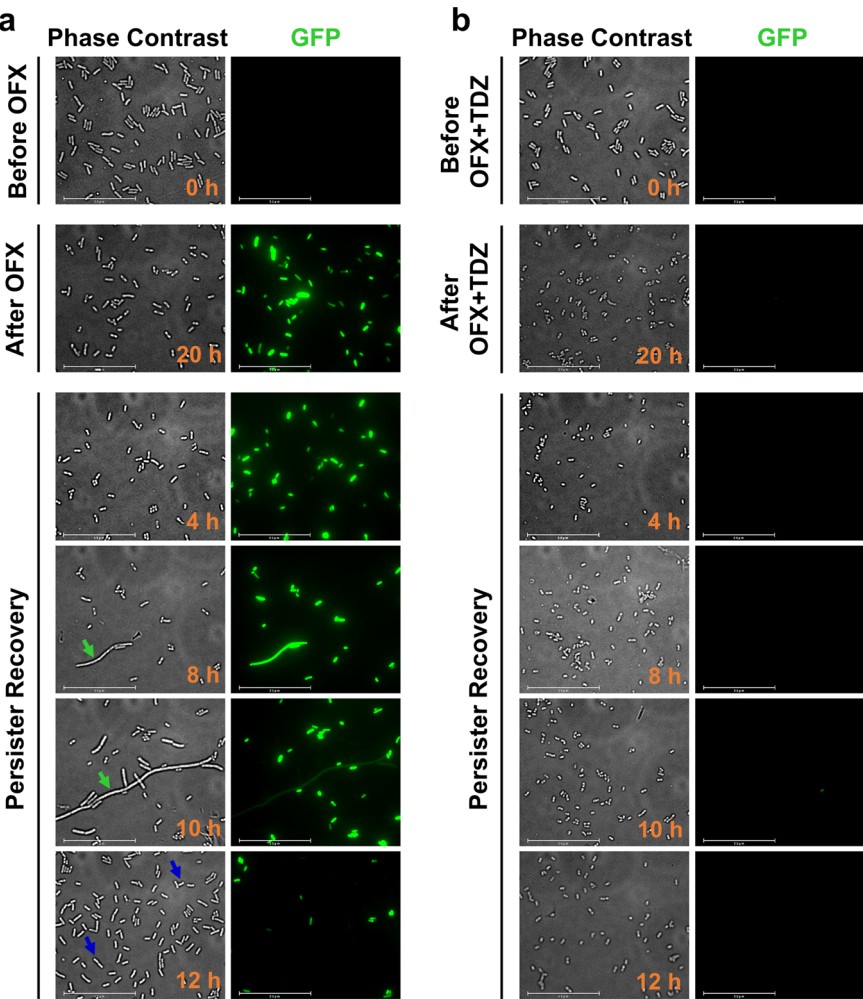

**Fig. 4 Phenothiazine drugs perturb OFX-induced RecA expression during recovery.** Early stationary phase *E. coli* MG1655 cells with P$_{recA}$-*gfp* (grown in LB) were treated with (**a**) OFX or (**b**) OFX + TDZ for 20 h. After treatment, cells were washed and transferred to liquid LB medium for persister recovery. At designated time points during persister recovery, 10 µl cells samples were spotted on 1% agarose pads for phase contrast and fluorescent microscopy. A representative image is shown; all independent biological replicates have similar results. Green arrow: elongated cells with high GFP, blue arrow: dividing exponential phase cells with low GFP. *n* = 3. Scale bar: 25 µm.

verifying the reproducibility of our data (Fig. 6b). Notably, we did not observe any growth deficiency in the mutant strains cultured in LB medium (Supplementary Fig. 11). Although the specific molecular interactions between phenothiazines and the identified proteins are yet to be defined, these data further support the ability of phenothiazine drugs to suppress cellular energy metabolism.

**Metabolic stimulation sensitizes stationary phase cells to OFX in the presence of phenothiazine drugs.** Given the effect of phenothiazine drugs on cellular membranes is transient (Fig. 6a), their mechanism of action may require active metabolic machineries. To determine cell growth phases for which phenothiazine drugs are more effective, samples were collected from 3 h to 24 h of cell growth (Supplementary Fig. 12). Samples were challenged with chemicals (OFX + phenothiazine) and persister cell levels were quantified. Persister levels in phenothiazine-treated cultures were very low at earlier time points (*t* = 3–6 h); however, their levels increased at later time points (*t* = 8–24 h) (Supplementary Fig. 12). Phenothiazine drugs are less effective for stationary phase cells, possibly because reduced metabolic activities in the bulk population make them intrinsically tolerant to OFX and phenothiazine drugs. We measured persister cell levels and *recA* expression following

treatment of cells with OFX during the transition to stationary phase (*t* = 5–9 h). In response to OFX, expression of *recA* was drastically reduced at later time points (Fig. 7a) and there was a negative correlation between *recA* expression and OFX persistence ($R^2$ = 0.7892; *P* < 0.0001) (Fig. 7b). Furthermore, the metabolic targets of phenothiazine drugs (i.e., cellular redox functions and ATP synthesis) were significantly reduced in the cells during the transition to stationary phase (Fig. 7c–e). There was a strong negative correlation between cellular metabolic activities (i.e., RSG staining, ATP synthesis, and *recA* expression) and cellular tolerance to treatments with OFX and phenothiazines ($R^2$ > 0.9; *P* < 0.0001) (Fig. 7f–h). These results indicate the inherent inactivity of the metabolic targets of phenothiazine or antibiotics in stationary phase cells accounts for the observed tolerance. Glycerol and glucose can be used efficiently by stationary phase cells[21,49] and they increase cellular electron transport chain activity and proton motive force[21]. Therefore, we determined whether metabolic stimulation with these carbon sources could enhance the killing of stationary phase cells. Stationary phase cells (*t* = 12 h), treated with OFX and/or the phenothiazine drugs 30 min after the addition of glucose or glycerol, showed an increase in ATP and redox activities (Fig. 7i–l), but OFX still induced DNA damage, as measured by *recA* expression (Supplementary Fig. 13a). Since OFX itself caused cell death at

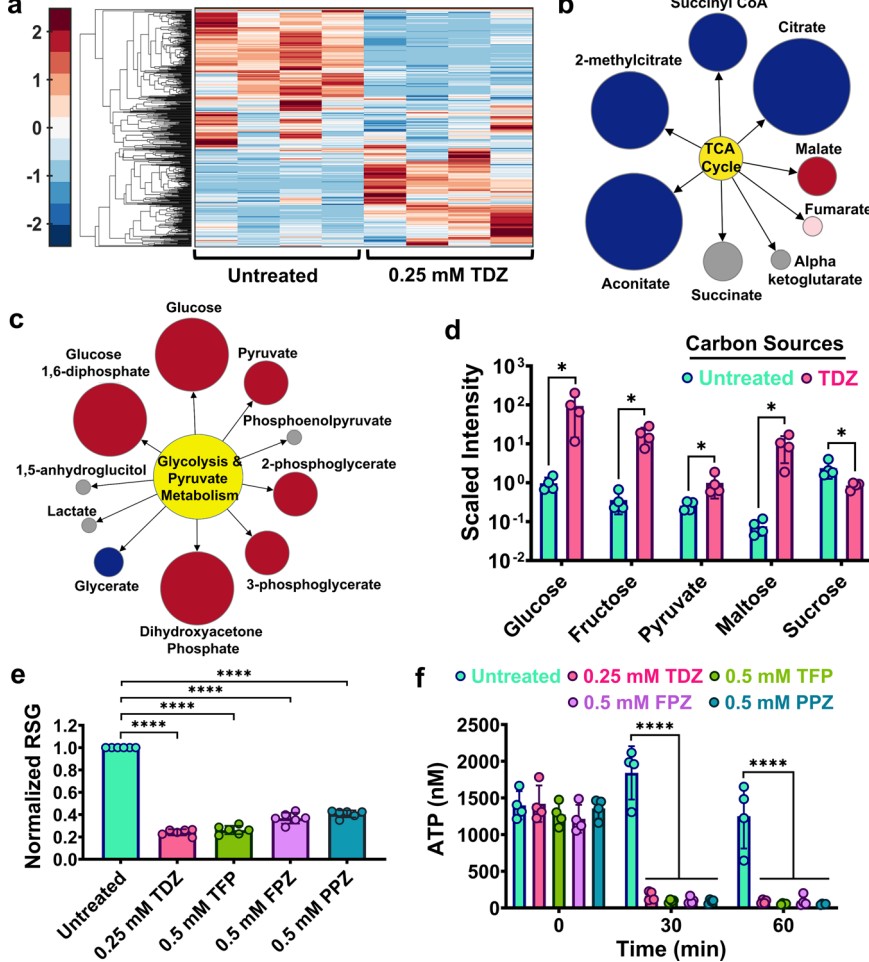

**Fig. 5 Phenothiazine drugs reduced energy metabolism in *E. coli* MG1655. a** Mass spectrometry analysis of TDZ-treated and control cells. *E. coli* MG1655 cells grown in LB were treated with 0.25 mM TDZ in early stationary phase ($t = 5$ h) for 20 h. After treatment, cells were collected and analyzed by mass spectrometry. Unsupervised hierarchical clustering was performed on the metabolic data. The data was standardized across all samples for each metabolite to obtain a mean value of 0 and a standard deviation of 1. Color codes show how a data point deviates from the mean value. Each column represents a biological replicate. $n = 4$. **b, c** Pathway enrichment maps. Metabolites of the TCA cycle and glycolysis from TDZ-treated cells were compared to those of untreated cells. The circle size is proportional to the ratio of the normalized intensities of metabolites between TDZ-treated and untreated cells. Blue ($P \leq 0.05$) and red ($P \leq 0.05$ for dark red; $0.05 < P < 0.10$ for light red) colors represent the metabolites that are significantly upregulated (red) or downregulated (blue) in the treatment group compared to the control. Gray indicates that there was no significant difference between the groups. $n = 4$. **d** Relative amounts of carbon source metabolites in the treatment and control groups obtained from mass spectrometry analysis. $n = 4$. **e** RSG staining of phenothiazine-treated cells. Early stationary phase cells ($t = 5$ h) were treated with phenothiazines for 20 h and then stained with RSG prior to analysis by flow cytometry. The mean fluorescence intensity of each treatment group was normalized to that of the control (untreated) group. $n = 6$. **f** ATP levels in phenothiazine-treated cells. Early stationary phase cells ($t = 5$ h) were treated with phenothiazine drugs at indicated concentrations and 100 µl of cell cultures were collected to measure ATP concentrations after 30 and 60 min. $n = 4$. Statistical analysis for panels **a**, **b**, **c**, and **d** was performed using Welch's t-test. Statistical analysis for panels **e** and **f** was performed using One-way ANOVA with Dunnett's post-test. *$P < 0.05$ and ****$P < 0.0001$. Data corresponding to each time point represent mean value ± standard deviation.

higher concentrations of the carbon sources[50,51] (Supplementary Fig. 13b), we chose a concentration (10 mM) that allowed OFX induction of DNA damage without a significant effect on cell viability (Fig. 7m and Supplementary Fig. 13b). The phenothiazine drugs killed OFX-damaged cells (Fig. 7m) but did not kill cells in the absence of OFX (Fig. 7m). Adding a carbon source also killed persisters in late-stationary phase cultures ($t = 24$ h) (Supplementary Fig. 13c), a notoriously challenging cellular state for the elimination of persisters.

**Phenothiazine drugs disrupt the proton concentration gradient across the cell membranes.** Although our results show phenothiazine drugs can transiently impair the cell membrane,

the effects of phenothiazine drugs are highly pleiotropic, as they can inhibit cellular redox activity, ATP formation, and transcription/translation (Figs. 4 and 5). Given that exponential phase cells are highly sensitive to phenothiazines (Supplementary Fig. 12), we successfully identified a concentration range of TDZ that eliminated OFX persisters (Fig. 8a–c) without affecting cellular membrane integrity in exponential phase cultures (Fig. 8a, b). This finding verifies that membrane integrity cannot be the only factor underlying the observed reduction in persister levels in co-treated cultures. If persistence is a dormant phenotype characterized by depressed metabolism, how can phenothiazine drugs eliminate persister cells? We think that persisters must still sustain a steady state adenylate energy charge to maintain cellular processes essential for their survival

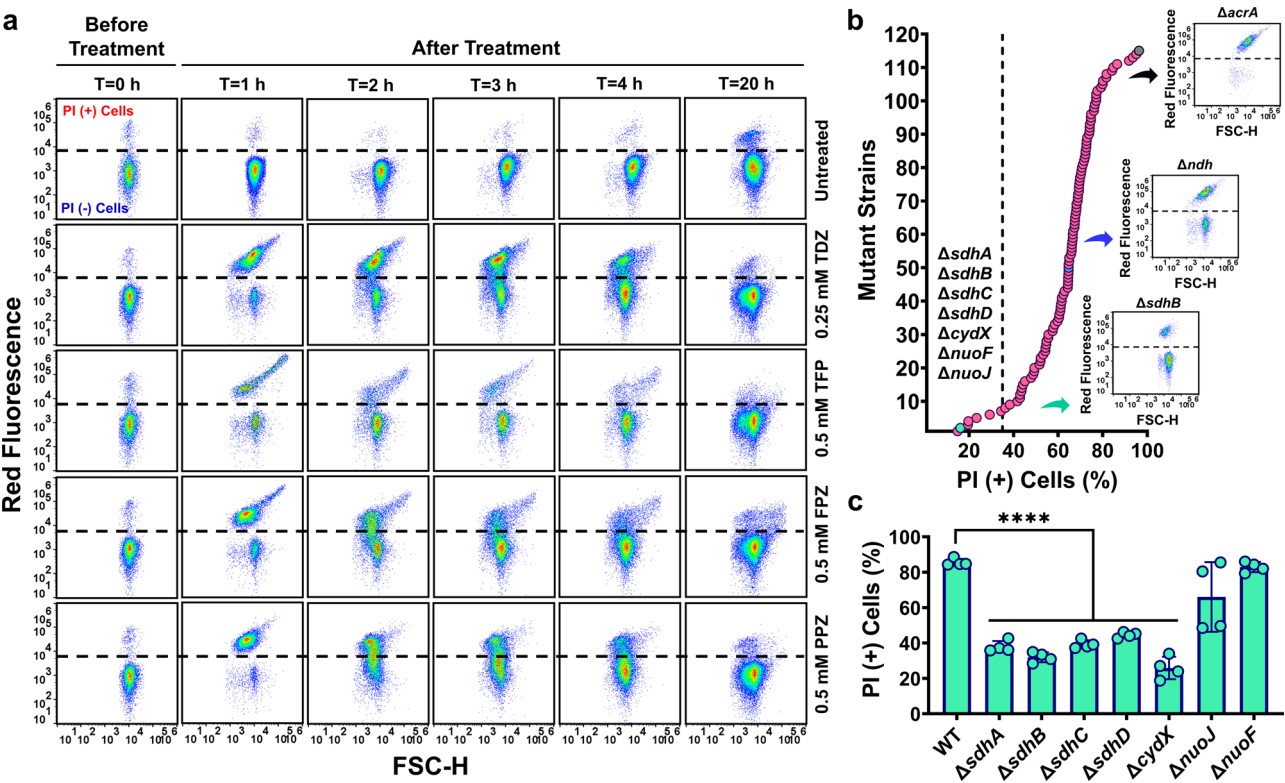

**Fig. 6 Phenothiazine can transiently permeabilize the cell membrane by inhibiting membrane-bound metabolic proteins. a** Phenothiazine treatment induced membrane permeabilization. *E. coli* MG1655 cells at early stationary phase (grown in LB) were treated with phenothiazine drugs at indicated concentrations. Following treatment, samples were collected and stained with PI at indicated time points for flow cytometry analysis. Live cells and ethanol-treated dead cells were used as negative and positive controls, respectively, to gate the PI negative (−) and PI positive (+) populations (see Supplementary Fig. 8b). *n* = 3. **b** Screening *E. coli* (K-12 BW25113) Keio Knockout Collection. Cells at early stationary phase (grown in LB) were treated with 0.25 mM TDZ for 1 h and then stained with PI for flow cytometry analysis to quantify PI (+) cell fractions (%). *n* = 1. **c** Phenothiazine-induced membrane permeabilization in *E. coli* MG1655 mutant strains. The mutant strains at early stationary phase were treated with 0.25 mM TDZ for 1 h and then stained with PI for flow cytometry analysis. *n* = 4. Statistical analysis was performed using one-way ANOVA with Dunnett's post-test. ****$P < 0.0001$. FSC-H: Forward scatter. Data corresponding to each time point represent mean value ± standard deviation.

(e.g., transcription/translation and DNA repair mechanisms)[26]. The pleiotropic effects of phenothiazines, which become detrimental to antibiotic-damaged persister cells, may be due to the ability of the phenothiazines to disrupt the proton motive force (PMF) of bacterial cells[46,47]. PMF, the driving force for ATP synthesis, has two components: the proton concentration gradient ($\Delta$pH) and electric potential gradient ($\Delta\psi$)[52], which bacterial cells should maintain for viability[53]. We postulate that sublethal concentrations of phenothiazines may prevent proton movement and disrupt the proton concentration gradient, $\Delta$pH, by inhibiting cytochromes and oxidoreductase complexes in *E. coli*. To verify this, we used an established methodology employing a membrane potential sensitive dye, 3,3′-dipropylthia dicarbocyanine iodide [DiSC$_3$(5)][53,54]. DiSC$_3$(5) self-quenches its own fluorescence when it accumulates in the cytoplasmic membrane of metabolically active cells. This fluorescence quenching is achieved when the cells have an intact membrane potential gradient, $\Delta\psi$[54]. If $\Delta\psi$ is disrupted, DiSC$_3$(5) is released into the extracellular environment resulting in an increase in fluorescence[54]. However, if $\Delta$pH is disrupted, an opposite effect should be observed, as live cells compensate for the $\Delta$pH disruption by increasing $\Delta\psi$[53]. When we treated metabolically active cells from exponential phase cultures with various inhibitors in the presence of DiSC$_3$(5), as described previously (see Methods)[53], we found sublethal concentrations of TDZ (Fig. 8a, b) dissipates $\Delta$pH (Fig. 8d). Polymyxin B was used as a control, as its polycationic peptide ring binds to a negatively charged site in the

lipopolysaccharide layer of the cellular membrane[55], resulting in the $\Delta\psi$ dissipation[53] (Fig. 8d). On the other hand, CPZ, the F1 complex inhibitor[56,57], was expected to disrupt $\Delta$pH (Fig. 8d). These results clearly explain the observed reduction in cellular ATP levels and redox activities shown in Fig. 5. Moreover, deletions of genes associated with cytochromes and oxidoreductase complexes significantly reduced OFX persister levels in exponential phase *E. coli* MG1655 cells (Fig. 8e), similar to phenothiazine-treated exponential phase cultures (Fig. 8c). A drastic reduction in persister levels was also obtained in *E. coli* MG1655 mutant strains in which the AtpA and AtpD subunits of F1 complex were deleted ($\Delta$atpA and $\Delta$atpD) (Fig. 8e). Altogether, our findings highlight the importance of metabolism for persister cell viability.

Finally, we treated cells with rifampicin (RIF, an inhibitor of transcription) or chloramphenicol (CAM, an inhibitor of translation) in the presence of OFX to determine whether these inhibitors can similarly eliminate OFX persisters. However, RIF and CAM could not eliminate OFX persisters across a wide range of tested concentrations (0–100 μg/ml) (Supplementary Fig. 14a), did not reduce cellular ATP levels (Supplementary Fig. 14e), and did not affect the cell membrane integrity like phenothiazine drugs (Supplementary Fig. 14d vs. Fig. 6a). Interestingly, our flow cytometry and microscopy analyses revealed RIF and CAM treatments reduced *recA* expression in the bulk cell populations (Supplementary Fig. 14b, c), though this reduction was not as drastically low as the phenothiazine-treated cultures (Supplementary Fig. 14b).

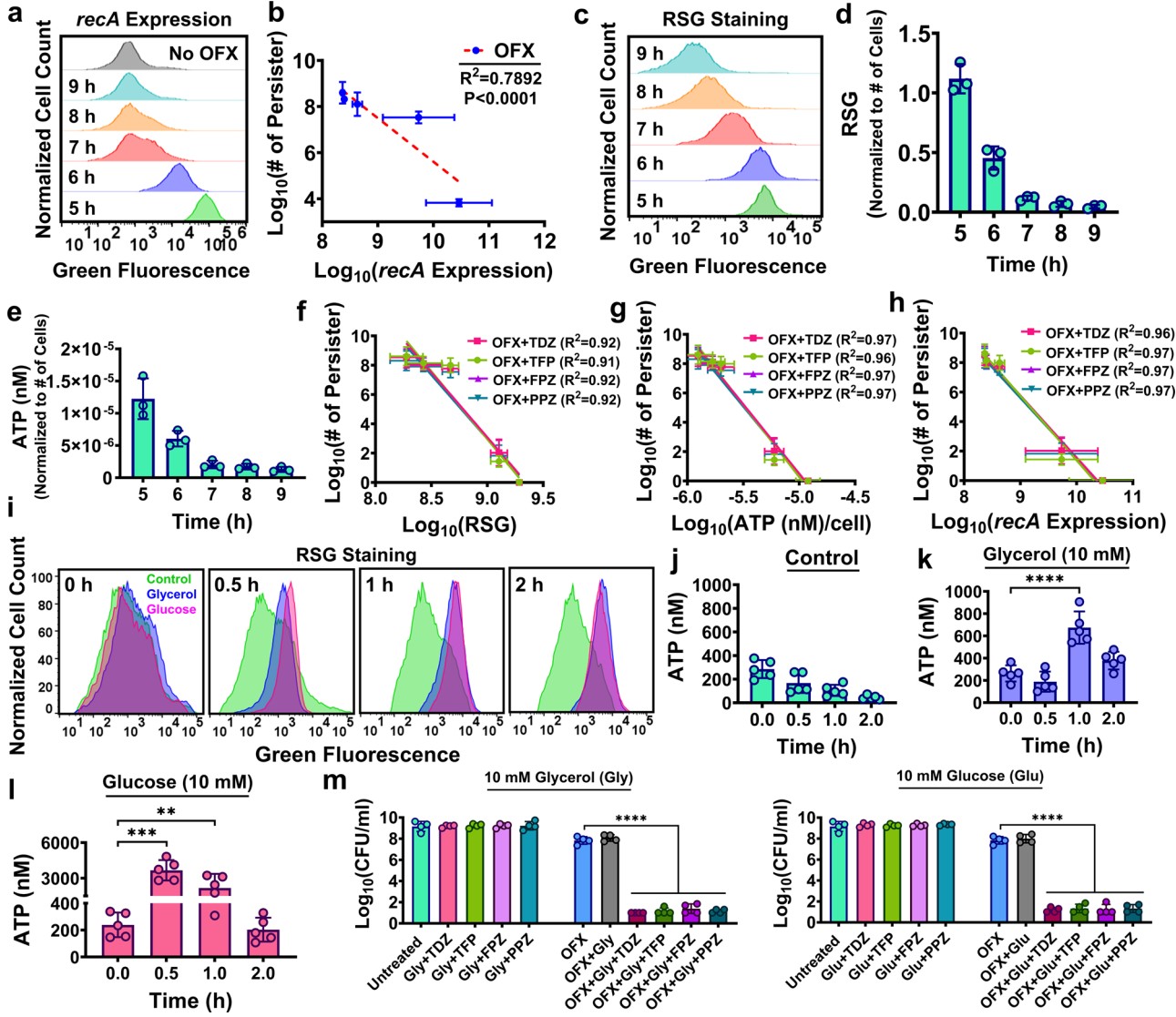

**Fig. 7 Metabolic stimulation can potentiate OFX and phenothiazine activities in stationary phase E. coli MG1655 cells. a** Induction of *recA* expression by OFX in stationary phase. *E. coli* cells with P$_{recA}$-*gfp* were treated with OFX at the indicated time points ($t = 5$–9 h). After a 20 h treatment, cells were analyzed by flow cytometry to measure GFP expression levels, and cells were collected and plated for viable cell counts. A representative flow cytometry diagram is shown. All independent biological replicates showed a similar trend. $n = 3$. **b** Correlation between persister levels and *recA* during stationary phase. A linear regression analysis between persister levels of OFX-treated cultures and *recA* expression was performed using the data obtained from panel a. $P < 0.0001$ (*F*-statistics). **c, d** RSG staining of stationary phase cells. *E. coli* MG1655 WT cells at indicated time points in the transition to stationary phase ($t = 5$–9 h) were stained with RSG dye to measure their redox activities by flow cytometry. A representative flow cytometry diagram is shown. All independent biological replicates showed a similar trend. $n = 3$. **e** ATP measurement of stationary phase cells. ATP concentrations were determined for cells at indicated time points in the transition to stationary phase ($t = 5$–9 h). $n = 3$. **f** Correlation between persister levels and cellular redox activities. A linear regression analysis between cellular redox activities (data from panel **d**) and persister levels of co-treated cultures (data from Supplementary Fig. 12) was performed, where $P < 0.0001$ (*F*-statistics). **g** Correlation between persister levels and cellular ATP levels. A linear regression analysis between cellular ATP levels (data from panel **e**) and persister levels of co-treated cultures (data from Supplementary Fig. 12) was performed, where $P < 0.0001$ (*F*-statistics). **h** Correlation between persister levels and *recA* expression. A linear regression analysis between cellular *recA* expression (data from panel **a**) and persister levels of co-treated cultures (data from Supplementary Fig. 12) was performed, where $P < 0.0001$ (*F*-statistics). **i–l** Metabolic stimulation of stationary phase cells with carbon sources. Mid-stationary phase *E. coli* MG1655 WT cells were supplemented with glycerol (10 mM), glucose (10 mM), or neither. At indicated time points, 100 µl of cell cultures were collected to measure their redox activities and ATP levels (nM). A representative flow diagram is shown. All independent biological replicates showed a similar trend. $n = 3$. **m** Survived cell levels in metabolically stimulated co-treated cultures. Cells at mid-stationary phase ($t = 12$ h) were treated with (i) OFX only, (ii) OFX + phenothiazine drugs, or (iii) phenothiazine drugs 30 min after adding glycerol or glucose to the cultures. After a 20 h treatment, cells were plated for viable cell counts. TDZ: 0.25 mM; TFP: 0.5 mM; FPZ: 0.5 mM; PPZ: 0.5 mM. $n = 4$. Statistical analysis was performed between the control and treatment group using one-way ANOVA with Dunnett's post-test. **$P < 0.01$, ***$P < 0.001$, and ****$P < 0.0001$. Data corresponding to each time point represent mean value ± standard deviation.

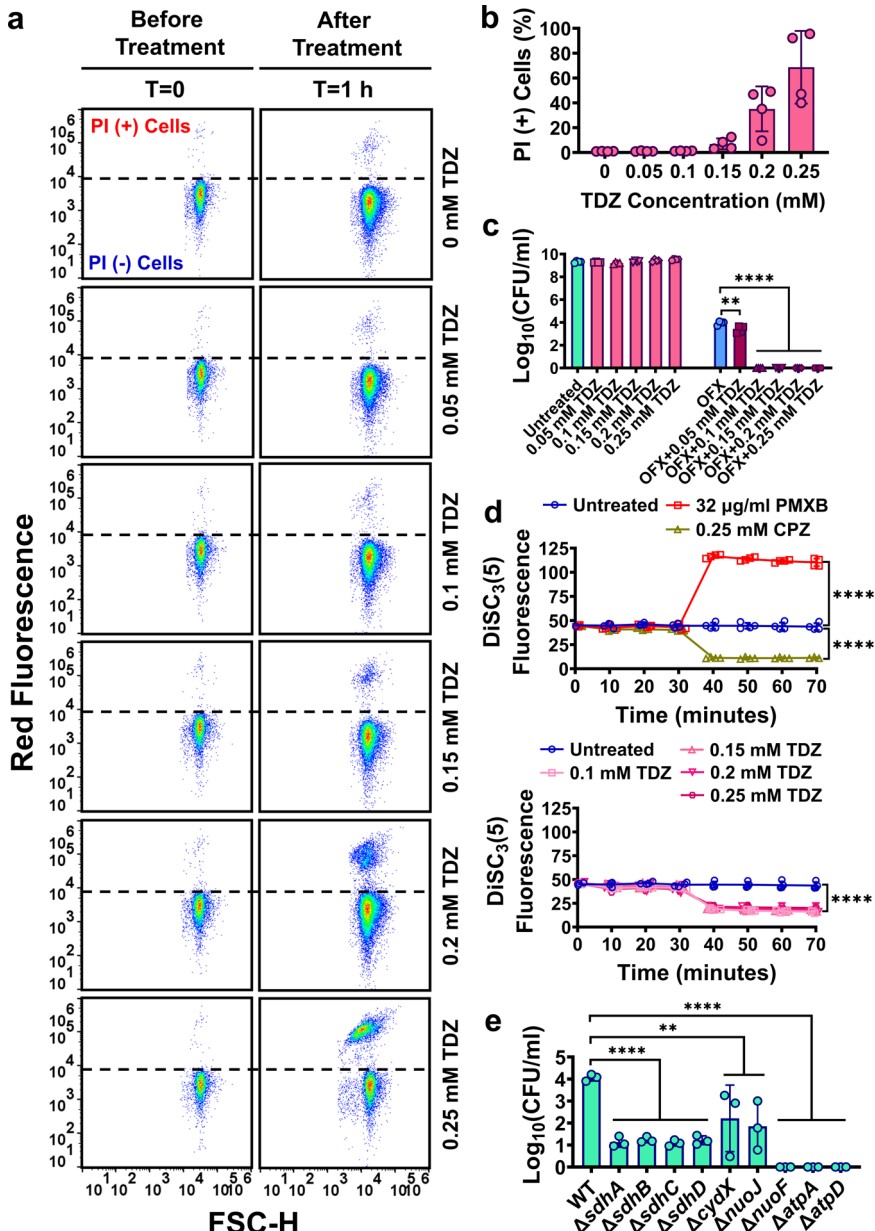

**Fig. 8 Phenothiazines dissipate the proton concentration gradient across the membranes of exponential phase cells. a** Phenothiazine-induced membrane permeabilization. Cells at exponential phase ($t = 3$ h) grown in LB were treated with TDZ at indicated concentrations. Cells before and after 1 h treatment were stained with PI for flow cytometry analysis to quantify PI (+) cell fractions. Ethanol-treated dead cells and untreated live cells were used as positive and negative controls, respectively (see supplementary Fig. 8b). $n = 4$. **b** PI (+) cell fractions (%) in TDZ-treated cell cultures. $n = 4$. **c** Persister enumeration. Cells at exponential phase ($t = 3$ h) were treated with OFX (5 μg/ml) and/or phenothiazines at indicated concentrations. After the 20 h treatment, cells were plated for viable cell counts. $n = 4$. **d** DiSC$_3$(5) assay. Metabolically active cells from exponential phase cultures ($t = 3$ h) were transferred to a buffer solution containing 1 μM DiSC$_3$(5) (see Methods). When the cells reached equilibrium, they were treated with chemicals (CPZ, TDZ, or polymyxin B) at indicated concentrations. At designated time points, samples were collected, and fluorescence levels were measured with a plate reader. Of note, the untreated control data is the same in both panels. Statistical analysis was performed between the final time points of control and treatment groups. $n = 4$. **e** Persister levels of *E. coli* MG1655 mutant strains. Mutant cells at exponential phase ($t = 3$ h) grown in LB were treated with OFX (5 μg/ml) for 20 h. After a 20 h treatment, cells were plated for persister enumeration. $n = 3$. Statistical analysis was performed using one-way ANOVA with Dunnett's post-test. **$P < 0.01$ and ****$P < 0.0001$. Data corresponding to each time point represent mean value ± standard deviation.

## Discussion

Here, we showed that phenothiazine drugs are highly effective anti-persister adjuvants as they can reduce persister populations obtained from various bacterial species (e.g., *P. aeruginosa*, *K. pneumonia*, and *A. baumannii*) or different antibiotic treatments (ofloxacin, ampicillin, ciprofloxacin, *etc.*). Although our chemical screening assay is based on the link between the SOS response and OFX persistence, the identified phenothiazine drugs indirectly inhibit DNA repair mechanisms. These drugs seem to inhibit a wide range of critical membrane-bound metabolic proteins, resulting in a downregulation of the PMF, ATP production, central carbon metabolism, and transcription and translational activities, which become greatly detrimental to antibiotic-damaged persister cells. However, these effects on persister populations in stationary phase become convoluted, as a significant number of cells from the stationary phase bulk population become intrinsically tolerant

to antibiotics. Although whether these stationary phase cells are true persister cells is still under debate[3], the effects of phenothiazines on persistence become clearer in exponential phase cultures, as antibiotics sterilize the bulk cell population. Further, our data indicate that the inhibition of transcription and translation by RIF and CAM, respectively, does not eliminate persister cells; however, the effects of RIF and CAM on cell metabolism and physiology are not as pleiotropic as those of phenothiazine treatments.

Persister cells are not necessarily a preexisting subpopulation of dormant cells[19] as they can be induced by antibiotics[11,12,58] and respond to antibiotic treatments[12,13]. Both persister cells and antibiotic-sensitive cells may show heterogeneity in their response to OFX[12,13]. Goormaghtigh et al. demonstrated some OFX persisters are metabolically active and proliferating prior to antibiotic treatment[13]. Metabolism is a highly complex network that can be involved in many aspects of persister cells. Although inhibiting the metabolism of exponentially growing cells may lead to dormancy and cell persistence[27,28], the stresses associated with active metabolism, such as intracellular degradation and the production of reactive oxygen species, can stimulate persister cell formation[22,59]. In addition to its importance in persister cell formation, metabolism is also important for persister survival and recovery[60] because the anabolic pathways and repair mechanisms for persister cell recovery and division require energy. This characteristic seems to be conserved among persister cells because we also observed a drastic reduction in ampicillin persisters following treatment with phenothiazine drugs. This finding might be expected, as cellular transcription/translation machineries are very active during the recovery of ampicillin persister cells (see Supplementary Fig. S3 in our previous study[61]).

The existence of an active energy metabolism does not necessarily suggest the upregulation of the energy metabolism. In our previous study, we used fluorescence activated cell sorting and RSG staining to demonstrate that persister cells in exponential phase cultures largely arise from the least-metabolically active subpopulation[19]. However, our follow up study revealed persisters cell phenotypes were largely populated in the most-metabolically active subpopulation in the stationary phase[22], a result in stark contrast to observations of persisters in exponential phase cultures. Although these two results seem contradictory at first glance, the flow cytometry diagrams from our RSG staining experiment clearly show the metabolism of the bulk population is significantly downregulated when the cells enter the stationary phase (Fig. 7c). In fact, persister cells may exist in a metabolic steady state, unlike the bulk population, which may explain why persister cells largely reside in the most-metabolically active subpopulation during the stationary phase. This may also explain why co-treatment (phenothiazine + OFX) is less effective for the bulk population in stationary phase cells. This metabolic model, which seems to be essential for persister cell survival, does not necessarily contradict the current prevailing paradigm, as persister cells may still have reduced metabolism when compared to the bulk population of exponential phase cells. Our continuous effort in the field verifies that inhibitors targeting cellular energy metabolism become highly detrimental for persister cells[22,39,59]. In addition, the effectiveness of PMF inhibitors against bacterial cells have recently been highlighted in various studies[53,62–64]. This may also explain the high hit rate of anti-persister adjuvants obtained from the small library used in the current study, as a considerable number of chemicals, whether approved by FDA or not, are known to disrupt the bacterial membrane bilayer or proteins essential for regulating membrane potential and energy metabolism[65]. However, whether these agents can truly establish a new way of treatment strategy for eradicating persister cells warrants further investigation.

Phenothiazine drugs are heterocyclic organic compounds with immense bioactivity and diverse applications in the fields of anesthesia, psychiatry, infectious diseases, and allergies[36,66]. These drugs are known to inhibit dopamine receptors in neurons and reduce psychotic symptoms[36,67]. Also, they are currently used to treat Parkinson's and Alzheimer's diseases[36,68]. Phenothiazine drugs can exhibit a wide range of antimicrobial activities when used in combination with conventional antibiotics. Specifically, they were shown to reduce efflux pump and cellular metabolic activities in a diverse range of bacterial species[40–44], which strongly supports our results. A key component of energy metabolism involves the oxidizing/reducing cellular compounds, which are the leading sources for ATP generation by the PMF. These components are produced by enzymes (mainly TCA cycle and ETC enzymes) that maintain the redox activities of cells (e.g., dehydrogenases). A number of independent research groups have already published that deletion of genes encoding these enzymes, including *ubiF*, *sucB*, *mdh*, *aceE*, *sdhC*, and *acnB*, drastically reduces persister levels[22,69–72]. Further, our study found no detectable persister cells in *E. coli* MG1655 mutant strains in which the AtpA, AtpD and NuoF subunits of critical metabolic proteins were deleted. We propose that depletion of ATP by phenothiazine drugs may further inhibit repair mechanisms because RecA catalyzes the DNA strand-exchange reaction during homologous recombination only in the presence of ATP[73,74]. Existing evidence supports this idea, as previous work used a biochemical assay to verify chlorpromazine, a phenothiazine analog, can bind to the F1 complex of ATP synthase[56,57], which can perturb the ΔpH component of PMF. Of note, phenothiazine-induced membrane permeabilization may not depend on ATP synthase, as the F1 complex is not membrane-embedded, thus explaining why our screening assay did not identify any F1 complex subunits. Also, TDZ has been shown to perturb the electric potential gradient (Δψ) of Gram-positive organisms by inhibiting NADH:quinone oxidoreductase II (NDH-II)[40,41,75]. Our screening assay indicates that the mutant strain lacking the *ndh* gene (encoding NDH-II) is still permeabilized by TDZ (Fig. 6b), and our PMF assay indicates that phenothiazines perturb the proton gradient of the PMF in *E. coli*, highlighting the existence of distinct mechanisms across species. Moreover, experimental differences, such as the timing of the addition of metabolic inhibitors, chemical concentrations, and the duration of the treatment with the inhibitors, as well as the existence of redundant interactions between the inhibitors and proteins likely affect the experimental outcomes.

In a previous study, we showed a correlation between persistence, degradation of cellular components (self-digestion), and stationary phase energy metabolism[22]. When cells are in a non-nutritive environment (e.g., a late-stationary phase culture), self-digestion generates energy, but it also causes cellular damage, which generates an antibiotic tolerant subpopulation of cells[22]. Previously, using a degradable GFP combined with high-throughput drug screening, we identified a few phenothiazine drugs that reduce both stationary phase protein degradation and persister formation[39]. Although we did not characterize the underlying mechanism in depth in our previous study, phenothiazine drugs likely inhibit stationary phase energy metabolism and stress factors associated with cellular metabolism[39]. With the use of untargeted metabolomics, a screening strategy for the mutant cell library, and assays to measure cellular redox activities, ATP levels, and the PMF, our present study demonstrates phenothiazine drugs perturb persister cell energy metabolism, which is essential for persister cell survival and recovery, and that co-treatment with antibiotics and phenothiazine drugs, a more convenient therapy, can kill recalcitrant Gram-negative bacterial persister cells.

## Methods

**Bacterial strains and plasmids**. *Escherichia coli* K-12 MG1655 wild type (WT), and pUA66 plasmids with genes encoding a green fluorescent protein (*gfp*) under the control of four promoters: P*recA*, P*recN*, P*sulA*, and P*tisB* were obtained from Dr. Mark P. Brynildsen at Princeton University (Supplementary Data 1)[12]. *E. coli* K-12 BW25113 WT and single deletions were obtained from Dharmacon Keio Collection (Dharmacon, Cat# OEC4988) (Supplementary Data 1). *E. coli* K-12 MG1655 Δ*recA*, Δ*sdhA*, Δ*sdhB*, Δ*sdhC*, Δ*sdhD*, Δ*cydX*, Δ*nuoJ*, and Δ*nuoF* were constructed in this study (Supplementary Data 2) using the Datsenko-Wanner method[76]. *Pseudomonas aeruginosa* (PAO1)[77] was a gift from Dr. Vincent H. Tam at the University of Houston (Supplementary Data 1). *Klebsiella pneumoniae* (CXY 130) and *Acinetobacter baumannii* (BAA-1605)[78] isolates were obtained from Dr. Kevin W. Garey at the University of Houston (Supplementary Data 1). An empty vector (pUA66 without the *gfp* gene) in *E. coli* was used as a control. *E. coli* MG1655 strains with pUA66 (low copy) or pQE-80L (high-copy) plasmids carrying an IPTG-inducible GFP, under the control of a synthetic T5 promoter, were obtained from the Brynildsen lab[22] and used to analyze the transcription/translation activities of the cells.

**Chemicals, media, and cultures conditions**. Unless noted otherwise, all chemicals were purchased from Fisher Scientific (Atlanta, GA), VWR International (Pittsburg, PA), or Sigma Aldrich (St. Louis, MO). *E. coli*, *K. pneumoniae*, and *A. baumannii* cells were grown in liquid Luria-Bertani (LB) medium. *P. aeruginosa* cells were grown in Mueller-Hinton (MH) medium[79]. LB agar medium was used to enumerate colony forming units (CFU) of *E. coli*, *K. pneumoniae* and *A. baumannii*; MH agar medium was used for *P. aeruginosa*[22,79]. LB broth was prepared by dissolving 5 g yeast extract, 10 g tryptone, and 10 g sodium chloride in 1 L deionized (DI) water. MH broth was prepared by dissolving 2.0 g beef extract powder, 17.5 g acid digest of casein, and 1.5 g soluble starch in 1 L DI water[39]. Agar media were prepared by dissolving premixed 40 g LB agar or 38 g MH agar in 1 L DI water, respectively[39]. Cells were washed free of antibiotics and other chemicals in phosphate buffered saline (PBS, 1X). Kanamycin (50 µg/ml) was added to the media for plasmid selection and retention[22]. To induce expression of genes for fluorescent proteins, 1.0 mM IPTG was added to the culture media[22]. Ofloxacin, ampicillin, norfloxacin, moxifloxacin, levofloxacin, and ciprofloxacin bactericidal antibiotics were used in persister assays[21,79,80]. Minimum inhibitory concentration (MIC) of antibiotics for different strains are tabulated in Supplementary Table 1. Two-fold macro-dilution method as described before was used to determine the MIC levels[39,81]. As persister cells survive high levels of antibiotics, the concentrations of the antibiotics used in persister assays were chosen to be much higher than the MIC levels (≥10x MIC) (Supplementary Table 1), consistent with previously published studies[21,79,80]. Phenotype MicroArrays (PM-11 to PM-25) containing 360 FDA-approved chemicals, small molecules, or antibiotics at four different concentrations, were purchased from Biolog Inc. (Hayward, CA). These arrays are preconfigured 96-well plates where chemicals have been dissolved in appropriate solvents and then dried in the wells. Chemicals identified from the PM-plate screening assays and antibiotics used for persister assays were tabulated with their vendor, catalog and purity information in Supplementary Table 2. Stock solutions for moxifloxacin (8 µg/ml) was prepared in DI water. Sodium hydroxide (0.01 N) was used to dissolve ofloxacin (5 µg/ml), norfloxacin (8 µg/ml), levofloxacin (3 µg/ml), and ciprofloxacin (3 µg/ml) in DI water. Stock solutions for thioridazine (TDZ; 0.1 M), trifluoperazine (TFP; 0.5 M), chlorpromazine (CPZ; 0.5 M), fluphenazine (FPZ; 0.5 M), amitriptyline (AMT; 0.5 M), potassium tellurite (PT; 0.1 M), polymyxin B (PMXB; 32 µg/ml), copper phthalocyanine-3,4′,4″,4‴′-tetrasulfonic acid tetrasodium salt (Cu-PcTs; 0.125 M), glucose (1.0 M), glycerol (1.0 M) and IPTG (1.0 M) were prepared in DI water. Stock solution for chloramphenicol (CAM; 50 mg/ml) was prepared in ethanol (100%). Stock solutions for perphenazine (PPZ; 0.5 M), hexachlorophene (HCP; 0.1 M), pentachlorophenol (PCP; 0.5 M), and rifampicin (RIF; 50 mg/ml) were prepared in DMSO. All chemical solutions were sterilized with 0.2 µm VWR syringe filters, except for those dissolved in DMSO. Solid and liquid media were autoclaved for sterilization. Overnight pre-cultures were prepared in a 14-ml Falcon test tube containing 2 ml liquid broth (LB or MH), inoculated from a 25% glycerol stock culture stored at −80 °C, and grown for 24 h at 37 °C with shaking at 250 revolution per minute (rpm). Experimental cell cultures were prepared by diluting the overnight pre-cultures (1:1000) into 25 ml fresh LB medium in 250-ml baffled flasks or 2 ml fresh LB medium in 14-ml test tubes. Cell cultures of bacterial species tested here generally reached the mid-exponential phase after about 3 h and early stationary phase after about 5 h (Supplementary Fig. 1). Cultures at t = 12 h and t = 24 h were considered mid- and late-stationary phase cultures, respectively (Supplementary Fig. 1). Unless otherwise stated, treatments were performed at early stationary phase by transferring 2 ml of culture to 14-ml Falcon tubes and then treating them with the indicated chemicals for 20 h. For untreated controls, solvents were added into assay cultures. Experimental details are explained below.

**Cell growth and persister assays**. Overnight pre-cultures were diluted 1:1000 in 25 ml LB medium in 250-ml baffled flasks at 37 °C with shaking at 250 rpm. Cell growth in the flasks was measured by optical density at 600 nm wavelength (OD$_{600}$) with a Varioskan LUX Multimode Microplate Reader (Thermo Fisher, Waltham, MA, USA). Plate reader data were collected using SkanIt Software V 5.0. Cell cultures (2 ml) at various time points in exponential and early-, mid- and late-stationary phase were collected from the flasks and transferred to 14-ml Falcon tubes for treatment with antibiotics or other chemicals at indicated concentrations and cultured with shaking at 37 °C for 20 h. After the treatment, a 1.0-ml sample from a Falcon tube was transferred to a microcentrifuge tube and centrifuged at 13,300 rpm (17,000x gravitational force or g). The 900-µl supernatant was removed and the cell pellet was washed three times with 900 µl PBS to reduce the antibiotic concentration to less than the MIC. After washing, the cells were resuspended in 100 µl PBS. A 10-µl sample of this cell suspension was serially diluted in 90 µl PBS in a round bottom 96-well plate and then 10 µl of the diluted cells were spotted on an LB agar plate to count the CFU. The remaining 90 µl of undiluted cell suspension was also plated to have a limit of viable cell detection equivalent to 1 CFU/ml. Persister levels were reported as the number of CFU per 1.0 ml of assay culture. The LB agar plates were incubated at least 16 h at 37 °C, as we found this incubation period was sufficient for growth of all viable *E. coli* cells. Initial cell count was determined prior to antibiotic treatment. The number of viable cells at various times during antibiotic treatment was determined by diluting 100 µl of culture in 900 µl PBS, then washing and plating the cells as described above. However, this procedure leads to a limit of viable cell detection of 10 CFU/ml.

***Pseudomonas aeruginosa, Klebsiella pneumoniae*, and *Acinetobacter baumannii* Persistence Assay**. Overnight cultures of *P. aeruginosa* PA01, *K. pneumoniae* (CXY 130) and *A. baumannii* (BAA-1605) strains were diluted 1:1000 in 2 ml MH (for *P. aeruginosa*) or LB broth (for *K. pneumoniae* and *A. baumannii*) in 14 ml Falcon tubes and grown at 37 °C with shaking (250 rpm). Early stationary phase (t = 5 h) cells were treated with OFX and/or the identified chemicals at indicated concentrations and grown at 37 °C with shaking for 20 h. Cells treated only with OFX or only with the drugs served as controls. After the treatments, 1.0 ml of culture from each tube was collected and washed, serially diluted and plated on MH agar or LB agar medium, as described above. The plates were incubated for 20 h at 37 °C, as this incubation period was sufficient for the colonies to grow.

**Fluorescent protein expression assay for reporter genes**. Overnight pre-cultures of *E. coli* MG1655 cells with reporter genes fused to the SOS gene promoters (P*recA*, P*recN*, P*sulA*, and P*tisB*) were diluted 1:1000 in 25 ml LB medium in 250 ml baffled flasks and grown at 37 °C with shaking. At early stationary phase (t = 5 h), cells were treated with OFX (5 µg/ml). After the treatment, 200 µl cultures were transferred to a flat-bottom 96-well plate to measure GFP using excitation and emission wavelengths of 485 nm and 511 nm, respectively, with a plate reader (Varioskan LUX Multimode Microplate Reader). Plate reader data were collected using SkanIt Software V 5.0.

**Chemical screening**. Overnight pre-cultures of *E. coli* cells with the P*recA*-*gfp* reporter were diluted 1:1000 in 25 ml LB medium in 250-ml baffled flasks and grown at 37 °C with shaking at 250 rpm. Early stationary phase (t = 5 h) cells were treated with OFX and immediately transferred to 96-well PM plates (100 µl cell culture/well) and grown at 37 °C with shaking. We used the *recA* reporter because the *recA* induction by OFX was the highest among the SOS genes tested. GFP levels were measured in the plate reader after 4 h of incubation. Wells that had only OFX (no chemicals) served as positive controls (GFP positive); wells that did not have OFX or chemicals served as negative controls (GFP negative). Chemicals that significantly reduced GFP expression in the presence of OFX compared with the positive controls were candidates for further testing. After the 20-h treatment, cells from the candidate wells were collected, washed with PBS twice and spotted on agar plates to enumerate persister cells. Similarly, survival of cells treated with candidate drugs without OFX was determined in order to identify chemicals that had bactericidal activities.

**Validation experiments for the identified chemicals**. Overnight cultures of *E. coli* cells with SOS gene reporters were diluted 1:1000 in 25 ml LB medium in 250-ml baffled flasks and grown at 37 °C with shaking at 250 rpm. Early stationary phase (t = 5 h) cells, treated or not with OFX, were immediately transferred to 14-ml Falcon tubes (2 ml cells/tube) followed by the addition of drugs at concentrations of 0–4 mM and grown at 37 °C with shaking. GFP expression was measured using 200 µl of treated cells in a flat-bottom 96-well plate in the plate reader. Cells treated only with OFX served as the positive control. Untreated cultures (no OFX and no chemicals) were the negative control. We also treated cells with the chemicals in the absence of OFX to determine whether they could induce the SOS response in the cells. Cells treated with OFX and/or drugs for 20 h were collected, washed and plated as described above to enumerate surviving cells.

**Persister recovery and microscope imaging**. A 2-ml culture of *E. coli* cells with the P*recA*-*gfp* reporter in a 14-ml tube was treated with OFX and/or TDZ in early stationary phase (t = 5 h). After 20 h, 1.0 ml of culture was washed with PBS three times by centrifugation at 13,300 rpm (17,000 × g) and pellets were resuspended in 1.0 ml LB medium. As high cell density may affect the persister recovery, the cell suspension was further diluted 1:5 in 2 ml LB medium in a 14-ml test tube and grown with shaking to allow for persister recovery. During the treatment and recovery periods, 10 µl of culture at indicated time points was collected and spotted

on a 1.0% (w/v) agarose pad on a microscope slide with a coverslip[19]. Fluorescent and phase contrast images of cells were taken using a microscope (EVOS FL Auto 2, Cat# AMAFD2000, Thermo Fisher) with a 100X (oil) objective (Olympus, AMEP4733, working distance = 0.3 mm) to determine the cellular morphology and GFP protein. For each time point, at least four phase contrast and GFP images were taken and more than 100 single cells were analyzed. Evos GFP light cube (AMEP4651) was used to capture the GFP images with 470/22 nm excitation and 510/42 nm emission wavelengths. Microscopy data were collected using Invitrogen EVOS FL Auto 2 software. The java-based image processing tool ImageJ was used for image analysis. To measure the CFU of recovering cells, 100 μl of cell cultures were collected at indicated time points, serially diluted and spotted on agar plates.

**PI staining**. Cells receiving the indicated treatments were diluted 1:100 in 1.0 ml PBS in flow cytometry tubes (5 ml round bottom Falcon tubes, size: 12 × 75 mm) to achieve the desired cell density (~$10^6$-$10^7$ cells/ml) for flow cytometry analysis. The resulted cell suspensions were treated with 20 μM PI dye. PI produces red fluorescence upon binding DNA; however, it can only penetrate cells with damaged membranes. The samples were vortexed briefly after adding the dye, and then incubated in the dark at 37 °C for 15 min before analyzing them by a conventional bench-top flow cytometer (NovoCyte 3000RYB, ACEA Biosciences Inc., San Diego, CA, United States). For flow cytometry analysis, we chose a slow sample flow rate (14 μl/ min) to have a sample stream diameter (i.e., core diameter) of 7.7 μm. The instrument has a constant sheath flow rate of 6.5 ml/min. The core diameter is determined by the ratio between the sample flow rate and the sheath flow rate. These conditions provide better data resolution for small cells such as E. coli. The flow cytometer utilizes low-power solid-state lasers. Cells were excited at a 561 nm wavelength and red fluorescence was detected with a 615/20-nm bandpass filter. At least 30,000 events were recorded for each sample. NovoExpress software was used to collect the data. PI stained dead cells, obtained after ethanol (70% v/v) treatment, were used as a positive control. PI stained live cells were used as a negative control. Forward and side scatter signals of untreated live cells were used to determine the cells on flow cytometry diagrams; the positive and negative controls were used to gate PI positive (+) and PI negative (-) cell populations (see Supplementary Fig. 8b).

**Transcription/translation activities of pUA66 and pQE-80L plasmids**. To assess the effect of the drugs on transcription/translation, we measured the amount of GFP produced by E. coli strains with the low-copy plasmid, pUA66, or the high-copy plasmid, pQE-80L, which have gfp under the control of a strong, IPTG-inducible T5 promoter. Overnight pre-cultures were diluted 1:1000 in 25 ml LB medium in 250 ml baffled flasks and grown with shaking at 37 °C. Early stationary phase (t = 5 h) cells were transferred to 14-ml Falcon tubes (2 ml cells/tube) and treated with drugs at the indicated concentrations followed by the addition of 1.0 mM IPTG. GFP was determined with the plate reader as described above. Cultures treated only with IPTG served as positive controls; cultures treated with or without chemicals in the absence of IPTG served as negative controls.

**Metabolomics**. Metabolites in thioridazine (TDZ)-treated E. coli MG1655 cells were analyzed by a Metabolon, Inc., facility (Morrisville, NC, USA). Cells at early stationary phase (t = 5 h) were treated with 0.25 mM TDZ and grown for 20 h at 37 °C with shaking (250 rpm). Cells were harvested by centrifugation at 4700 rpm at 37 °C for 15 min to collect a pellet of about 100 μl (~$10^{10}$ cells). The cells were washed once with PBS by centrifugation (13,300 rpm, 3 min at 4 °C), and then, the cell pellets were frozen in an ethanol/dry ice bath for 10 min. Control cells were prepared similarly but without TDZ treatment. The extracts from treatment and control groups were processed using ultra-high-performance liquid chromatography-tandem accurate MS to quantify a wide range of metabolites. Sample extraction, preparation, instrument settings, and conditions for the MS platform were performed according to Metabolon's protocols (see Supplementary **Method** for details), as described elsewhere[82]. The data were compared with an extensive metabolite library (standards) from Metabolon to identify true metabolites in the samples and to remove false positives arising from instrument noise, process artifacts, and redundant ion features (see Supplementary **Method**). The biochemical data were normalized to protein concentration (assessed by the Bradford assay). Welch's two-sample t-test was used to identify metabolites that differed significantly between the control and treatment groups.

**Redox sensor green dye staining**. Redox Sensor Green (RSG) dye (Thermo Fisher, Catalog# B34954) measures bacterial reductase activity. Cells at early stationary phase (t = 5 h) were treated with or without drugs for 20 h and diluted 1:100 in 1.0 ml PBS containing 1.0 μM RSG dye in flow cytometry tubes. After brief (~10 s) vortexing, the samples were incubated in the dark at 37 °C for 10 min followed by flow cytometry analysis. We used the same flow cytometry method described above (see the "PI staining" section); however, cells were analyzed with a laser emitting light at 488 nm and the green fluorescence was detected with a 530/30 bandpass filter. For control, cells were treated with 10 μM carbonyl cyanide m-chlorophenyl hydrazone (CCCP) before adding RSG (Supplementary Fig. 8a and Supplementary Fig. 9).

**ATP measurement**. BacTiter-Glo™ Microbial Cell Viability Assay kit (Catalog# G8230, Promega Corporation, Madison, WI, USA) was used to measure the ATP concentrations. After treatment of the cells with drugs, 100 μl of culture was mixed with 100 μl of luciferase solution and incubated at room temperature for 5 min. Luminescence was measured in a plate reader with LB medium as a control for background luminescence. Standard curves were prepared using rATP (Promega Corporation, catalog# P1132) dissolved in LB. Plate reader data were collected using SkanIt Software V 5.0.

**Screening E. coli (K-12 BW25113) keio knockout collection**. Overnight cultures of single mutants (Supplementary Data 1) carrying kanamycin resistance genes were diluted 1000-fold in 14-ml test tubes containing 2-ml LB medium and grown at 37 °C with shaking (250 rpm). Kanamycin (50 μg/ml) was added in overnight and treatment cultures to prevent contamination. When the cultures reached the early stationary phase (t = 5 h), cells were treated with TDZ (0.25 mM) for 1 h. Treated cells were then diluted (1:100) in 0.85% NaCl solution to achieve the desired cell density (~$10^6$ cells/ml). The resulted cell suspensions were stained with 20 μM PI and incubated in a dark for 15 min at 37 °C. Stained cells were analyzed with a flow cytometer using the conditions described above (see the "PI staining" section). Cells were excited at 561 nm wavelength and red fluorescence was detected with a 615/20 nm bandpass filter. At least 30,000 cells were analyzed for each sample. Forward and side scatter parameters of unstained live cells were used to determine the cells on the flow diagram. Ethanol (70%v/v) treated, PI stained dead cells served as the positive control. PI stained live cells served as the negative control. These controls were used to gate the PI (+) and PI (−) cell populations, respectively (see Supplementary Fig. 8b). Validation experiments for the identified genes were performed applying the same PI staining method for E. coli MG1655 single gene deletions.

**Metabolic stimulation of stationary phase cells**. Glycerol and glucose were used to stimulate the metabolism of intrinsically tolerant and metabolically repressed stationary phase cells[49]. Cell cultures at t = 11.5 h or t = 23.4 h were transferred to 14 ml Falcon tubes (2 ml cells/tube), supplemented with glycerol and glucose (concentrations: 0, 10, 25, 50 mM), grown with shaking at 37 °C for 30 min, and then treated with OFX and/or candidate drugs at 12 h (mid-stationary phase) or 24 h (late-stationary phase). Cells treated only with OFX or only with the drugs served as controls. After the 20-h treatment, cultures were washed, serially diluted and plated to count persister cells. Measurements for ATP, redox activities and RecA (as measured by GFP) expression in metabolically stimulated stationary phase cells were performed in the absence of antibiotics and drugs, as described above.

**DiSC$_3$(5) assay**. Overnight cultures of E. coli MG1655 WT cells were diluted 1000-fold in 14-ml test tubes containing 2-ml LB medium and grown at 37 °C with shaking (250 rpm). Mid-exponential phase (t = 3 h) cells were collected and washed with a buffer (5 mM HEPES and 20 mM Glucose in DI water)[53] three times by centrifugation. Finally, cells were diluted in the buffer to obtain $OD_{600}$~0.1 and stained with 1 μM DiSC$_3$(5) and incubated in a dark place at 37 °C. Fluorescence levels of the cells were measured with a plate reader every 10 min. Once the cells reached an equilibrium state, cultures were treated with chemicals (CPZ, TDZ and polymyxin B) at indicated concentrations and incubated in a dark place at 37 °C. At designated time points, samples were collected, and fluorescence levels were measured with a plate reader. Cells were excited at a 620 nm wavelength and fluorescence was detected with a 670-nm emission filter. Plate reader data were collected using SkanIt Software V 5.0.

**Persister quantitation in E. coli MG1655 single gene deletions**. Overnight cultures of mutant strains were diluted 1000-fold in 14-ml test tubes containing 2-ml LB medium and grown at 37 °C with shaking (250 rpm). Exponential phase (t = 3 h) cells were then treated with OFX for 20 h. Samples before and after OFX treatment were collected, washed and plated on LB agar for the CFU enumeration.

**Statistics and reproducibility**. Biphasic kill curves were generated using a non-linear model[7,39]:

$$Log_{10}n_i = Log_{10}\{(n_0 - p_0)e^{-k_n t} + p_0 e^{-k_p t}\} \qquad (1)$$

where, $n_0$ = initial number of cells (CFU/ml), $p_0$ = initial number of persister cells (CFU/ml), $k_n$ = killing rate of normal cells, $k_p$ = killing rate of persister cells, $t$ = time (h), and $n_i$ = number of survived cells (CFU/ml) at t. A linear regression analysis was performed to determine the correlation coefficients. Both linear and non-linear regression analyses were evaluated with F-statistics[7,39]. Metabolomics data was analyzed by Welch's two-sample t-test to identify metabolites that differed significantly between control and treatment groups[83]. One-way ANOVA with Dunnett's post-test was used for all other pairwise comparisons[39]. A minimum of three independent biological replicates (unless otherwise stated) were performed for all experiments. In all figures (except flow diagrams), data corresponding to each time point represent mean value ± standard deviation. For statistical significance analysis the threshold value of P was set as *P < 0.05, **P < 0.01,

***P < 0.001, and ****P < 0.0001. All figures were prepared using GraphPad Prism 9.3.0. The statistical analyses were performed using GraphPad Prism 9.3.0 statistical functions. Metabolomics data was clustered using the "Clustergram" function of MATLAB (V R2020b). FlowJo V 10.7.1 was used to analyze the data obtained from flow cytometry.

## Data availability

The raw data supporting the conclusions of this article are available in FigShare. https://doi.org/10.6084/m9.figshare.17284823[84]. All other data are available from the corresponding author on reasonable request.

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

## Acknowledgements
The authors would like to thank the members of Orman Lab for their help. This study was supported by NIH/NIAID K22AI125468 Career Transition Award, NIH/NIAID R01-AI143643 Award, and the University of Houston start-up grant.

## Author contributions
S.G.M., T.V.N., and M.A.O. conceived and designed the study. S.G.M. and T.V.N. performed the experiments. S.G.M., T.V.N., and M.A.O. analyzed the data and wrote the paper. All authors have read and approved the manuscript.

## Competing interests
The authors declare no competing interests.

## Additional information

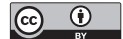

