## [Peer Review File · Communications Biology]

Reviewers' comments:

Reviewer #1 (Remarks to the Author):

Pleiotropic actions of phenothiazine drugs are detrimental to persister cells in bacteria

This paper presents a comprehensive study on the potential for a class of compounds, phenothiazine drugs, to act in combination with fluoroquinolone antibiotics to eradicate bacterial persister cells.

The phenothiazines were identified, along with several other types of compounds, through a small screen of 360 compounds. They were selected for further studies as they are already approved antipsychotic drugs, which gives this paper particular importance as potential beneficial effects on antibiotic therapy could be potentially be tested in the clinic relatively easily.

A range of studies are conducted to decipher how the phenazines work. Importantly, persister cells from various stages of culture growth are studied.

The effect on persister cells is demonstrated to work with multiple members of the phenazine class, and was demonstrated to work with another antibiotic class – ampicillin.

While most studies were conducted in *E. coli*, the persister reduction was also shown to work in *Pseudomonas aeruginosa*.

However, the significance of the paper is reduced as a similar study was published by the authors (*Front. Microbiol.*, 27 March 2020 | <https://doi.org/10.3389/fmicb.2020.00472>) where a library was screened and chlorpromazine identified to reduce persistence in *Escherichia coli*. and *Pseudomonas aeruginosa*. Effects on metabolism were also studied. The authors need to clearly differentiate what is new in this paper.

One area that would further strengthen the paper would be similar experiments to indicate if the effect also exists in the other key Gram-negative pathogens e.g. *Acinetobacter baumannii* and *Klebsiella pneumoniae* – and to confirm that it extends across other members of the fluoroquinolone class e.g. ciprofloxacin, delafloxacin - it is recommended that this is done.

Given the previous *Frontiers Microbiol* article, I believe that to increase the significance to a level suitable for *Communications Biology*, an *in vivo* study is required to demonstrate that the combination is more effective at clearing infections.

Other

- 1) given the high hit rate from a small library, comment on the likelihood of being able to identify other compounds
 - 2) purity and catalog numbers of the antibiotics should be reported in Methods section, the purity of the chemicals in Table S4 should also be reported
 - 3) p4: comment on purchased chemical library – how do you conduct the screen if you don't know the concentrations of the drug? It seems very odd this is not supplied for a purchased library.
 - 4) p 11: full name of DiSC3(5)] incorrect – no '0' or space after thia
 - 5) p19 – no details on flow cytometry conditions provided, what x g for centrifugation?
- Figure 1: suggest splitting this into 3 Figures as the 'wall of graphs' is difficult to comprehend and each graph is too small. Separate into drug screening (A-D), phenazine characterisation (E-I) and the *Pseudomonas*/time dependence (J-L).
- Fig 4: again, very difficult to see. Potentially make figure full page, have timepoints run vertically rather than horizontally

Reviewer #2 (Remarks to the Author):

The ms by Mohiuddin et al aims at identifying chemicals that are eradicating persister cells to ofloxacin, a fluoroquinolone inducing the SOS response.

The authors screened a library of compounds first on their ability to inhibit the SOS response and then refined their criteria: reduction of the number of persister cells and low toxicity. They ended

up with 7 candidates, among which 3 are phenothiazine drugs. They also tested 2 additional FDA-approved phenothiazine drugs. They follow-up their study only with phenothiazine compounds. Finding anti-persisters is certainly timely, and to my knowledge it is the first example of FDA-approved compounds. However, how in vitro persistence assays recapitulate persistence in a host is still unknown, especially for intracellular pathogens.

While the work seems promising, I found the ms difficult to follow. The rationale is not always clear, a lot of figures and supplementary figures are presented, and at the end the reader (at least I) get lost in the twists and turns of the story.

Major comments:

1. I would suggest presenting the data for only one or 2 drugs (TDZ?) as main figures. This would lighten the number of panels in each figure. The others could be shown in Sup.
2. The authors should consider the time of treatment as 0h and not 5h (of culture) in all their graphs and they should mention the culture medium used. The legends are not always clear.
3. While the authors claim that they screened for drugs inhibiting the SOS response (tested with a variety of transcriptional reporters), they also show that these drugs are inhibiting translation in general. Moreover, these drugs also eradicate persister cells to a totally unrelated antibiotic, ampicillin. This indicate that these drugs are not specific to ofloxacin-induced repair mechanisms – this should be stated clearly and linked with the other effects of the drugs (membrane depolarization, ATP drop, etc). Basically, inhibition of translation appears to a side effect of the metabolic problems as transcription and/or translation inhibition on the contrary increase persistence frequency. The authors should comment on that.
4. Metabolomics data show that TDZ impairs TCA cycle and ATP production and leads to accumulation of metabolites associated with glycolysis and carbon metabolism. However, in the same conditions, TDZ and other phenothiazine drugs do not impair survival. How is this possible? What are the effects on growing cells? It would be interesting to show a growth curve in the presence of TDZ.
5. Fig 4D: The authors show that post-treatment with TDZ do not affect persistence frequency to ofloxacin. If TDZ impairs the recovery of persisters by decreasing the metabolism, I suppose that adding TDZ after ofloxacin should also inhibit persister recovery. How do the authors explain this observation? Does ofloxacin impair the uptake of phenothiazines? Could this be evaluated experimentally?
6. Phenothiazine drugs have no effect on persistence of cells grown in stationary phase (6h of culture and more). What is the explanation? Do phenothiazines penetrate stationary phase cells? The authors should comment on that and test whether these drugs inhibit persistence of cells in log phase.
7. Fig 5A: PI permeabilization induced by phenothiazines is transient. What is the explanation? Does that imply that membrane depolarization is not the main effect of these drugs leading to increase persistence? If it was the case, persistence should increase with time? What is happening in cells treated with both ofloxacin and phenothiazines?
8. The rationale behind the experiments presented in Fig 5B is unclear (PI permeability of KO mutants).
9. Fig 6C: It seems that some bar duplications have occurred. The blue bars represent the TDZ treatment at different concentrations. As far as I can see, bars at 0.1mM, 0.15mM and 0.2mM are identical and those at 0mM and 0.25mM as well. It seems to me that the data should be presented in one single graph as untreated and ofloxacin controls appear to be the same for the different conditions.
10. Figure 6D, E, F and related text are not clear. The text seems to be rather a discussion than what is needed to introduce the rationale of the presented experiments. Fig 6D appears to be already published? Is that correct? Did the authors redo the experiment in the conditions used in this ms?
11. L366: Not sure why the authors postulate that depletion of ATP inhibits repair. Their data show that translation is inhibited by phenothiazines, implying that repair proteins are not synthesized. This is observed in Fig4.
12. At the end, what are the main conclusions about the mechanisms of action of those drugs? Why are the effects only revealed in the presence of ofloxacin? What is happening in the presence of ampicillin? Why the effect of phenothiazines is different in cells grown for 5h (early stationary phase) and grown longer? Does that imply that persisters arising from growing cells and stationary

phase cells are different? Does it suggest that the persistence mechanisms are different? Are stationary phase 'true' persisters or 'just' stationary phase cells that are refractory to a lot of different stress – antibiotics, phenothiazines? By which mechanisms could the synergy between ofloxacin and phenothiazines be explained? What is(are) the mechanisms behind the need to add both compounds at the same time to observe an effect of phenothiazines on persistence frequency? The discussion section should address all these points and put the observations in perspective with the current knowledge regarding persistence. I also suggest to improve the introduction by providing a more general view of the mechanisms involved in persistence and not to focus only on the SOS response as it seems very unlikely that phenothiazines specifically target DNA repair.

Minor comments:

Even though 'resuscitation' is used by some groups in the persistence field, I think it is not appropriate in a scientific context.

Maybe original papers instead of too many reviews should be cited in the introduction

L44: I don't understand what the authors mean by 'persisters are capable of physiological responses to external factors that are essential for resuscitation after antibiotic removal'. Should be more explicit.

Reviewer #1 (Remarks to the Author):

Pleiotropic actions of phenothiazine drugs are detrimental to persister cells in bacteria

This paper presents a comprehensive study on the potential for a class of compounds, phenothiazine drugs, to act in combination with fluoroquinolone antibiotics to eradicate bacterial persister cells.

The phenothiazines were identified, along with several other types of compounds, through a small screen of 360 compounds. They were selected for further studies as they are already approved antipsychotic drugs, which gives this paper particular importance as potential beneficial effects on antibiotic therapy could be potentially be tested in the clinic relatively easily. A range of studies are conducted to decipher how the phenazines work. Importantly, persister cells from various stages of culture growth are studied.

The effect on persister cells is demonstrated to work with multiple members of the phenazine class, and was demonstrated to work with another antibiotic class – ampicillin. While most studies were conducted in *E. coli*, the persister reduction was also shown to work in *Pseudomonas aeruginosa*.

Response: We thank the reviewer for their constructive feedback.

However, the significance of the paper is reduced as a similar study was published by the authors (Front. Microbiol., 27 March 2020 | <https://doi.org/10.3389/fmicb.2020.00472>) where a library was screened and chlorpromazine identified to reduce persistence in *Escherichia coli*. and *Pseudomonas aeruginosa*. Effects on metabolism were also studied. The authors need to clearly differentiate what is new in this paper.

One area that would further strengthen the paper would be similar experiments to indicate if the effect also exists in the other key Gram-negative pathogens e.g. *Acinetobacter baumannii* and *Klebsiella pneumoniae* – and to confirm that it extends across other members of the fluoroquinolone class e.g. ciprofloxacin, delafloxacin - it is recommended that this is done. Given the previous Frontiers Microbiol article, I believe that to increase the significance to a level suitable for Communications Biology, an in vivo study is required to demonstrate that the combination is more effective at clearing infections.

Response: We have now added more details about our previous study in the current manuscript to clarify this issue (see lines 151-159 and 417-430). We previously showed that pretreatment of stationary phase Escherichia coli cells with thioridazine (TDZ), trifluoperazine (TFP), or chlorpromazine (CPZ) significantly reduced type I persisters¹, which are typically formed by their slow exit from stationary phase in fresh media². Although we did not demonstrate the underlying mechanism, we speculated that phenothiazine drugs reduced type I persister formation by inhibiting stationary phase metabolism¹. Phenothiazine drugs are thought to interact with a diverse range of membrane-bound proteins³⁻⁷; however, the specific mechanisms linking these drugs to bacterial persistence remain unknown. Because of their potential therapeutic value, we performed a series of labor-intensive experiments (e.g., untargeted metabolomics, a screening strategy for the mutant cell library, and assays to measure cellular redox activities, ATP levels, and the proton motive force) to gain a better understanding of their anti-persister characteristics. These drugs disrupt the cellular proton motive force by dissipating the proton concentration gradient (ΔpH) across the cell membrane of *E. coli* cells, resulting in a downregulation of the PMF, ATP production, central carbon metabolism, and transcription and translational activities. The drugs at the concentrations tested can transiently perturb

cellular membrane integrity of *E. coli* cells while the cells can still grow in the presence of these drugs (see new Supplementary Fig. S10). *E. coli* mutant strains that were less permeabilized by phenothiazines generally lack genes encoding protein subunits of succinate: quinone oxidoreductase, cytochrome *bd-I* ubiquinol oxidase and NADH:quinone oxidoreductase complexes. Moreover, a drastic reduction in persister levels was also obtained in *E. coli* mutant strains in which the *AtpA* and *AtpD* subunits of F1 complex were deleted (Δ *atpA* and Δ *atpD*). All of these research elements significantly contributed to the novelty of this study.

Although we were not able to perform animal studies due to the lack of resources in our laboratory, we performed all other experiments requested by the reviewer. In the current study, we showed that treatment with phenothiazine and norfloxacin, moxifloxacin, levofloxacin, and ciprofloxacin eliminated quinolone persister phenotypes in early stationary phase *E. coli* cultures (See Fig. 1, which corresponds to Fig. 2d in the manuscript).

Fig. 1. Cell survival after quinolone and phenothiazine treatment. Cells (*E. coli* MG1655) treated with indicated quinolone at 10xMIC and/or phenothiazines for 20 h were plated for viable cell counts. NOR: norfloxacin (0.8 μ g/ml); MXF: moxifloxacin (0.8 μ g/ml), LVX: levofloxacin (0.3 μ g/ml); CIP: ciprofloxacin (0.2 μ g/ml). N=4. Statistical analysis was performed using one-way ANOVA with Dunnett's posttest, where * $P < 0.05$, *** $P < 0.001$, **** $P < 0.0001$.

Phenothiazine also reduced ofloxacin persister levels of pathogenic organisms *Klebsiella pneumoniae* and *Acinetobacter baumannii* (Fig. 2, which corresponds to Fig. 3 in the manuscript). We believe that these additional results further strengthen our paper.

Our work also provides evidence in support of the conclusion that altering the bacterial metabolic state can increase antibiotic susceptibility; the importance of this topic has been discussed extensively in a review article⁸. Persister metabolism is one of the most controversial topics in the persister research field. Although persister cells were shown to have reduced metabolism when compared to the bulk population of exponential phase cells, persister cells still need to sustain a steady-state adenylate energy charge to maintain cellular processes essential for their survival (e.g., transcription/translation, DNA repair mechanisms). We need a clear and much-needed picture of this controversial topic, and our study will significantly contribute to

this area. These points have extensively discussed in the manuscript now (see the Discussion section).

Fig. 2. Phenothiazine drugs reduced persister levels of Gram-negative bacteria. Early stationary phase cells of *Klebsiella pneumoniae* and *Acinetobacter baumannii* strains (see Supplementary Fig. S1 for growth curves) were treated with OFX at 10xMIC (25 μ g/ml and 140 μ g/ml, respectively) and the phenothiazine drugs for 20 h. Then, cultures were plated for viable cell counts. Statistical analysis was performed using one-way ANOVA with Dunnett's posttest, where **** $P < 0.0001$.

Other

1) given the high hit rate from a small library, comment on the likelihood of being able to identify other compounds

Response: Our continuous effort in the field verifies that inhibitors targeting cellular energy

metabolism become highly detrimental for persister cells^{1,9,10}. In addition, the effectiveness of PMF inhibitors against bacterial cells have recently been highlighted in various studies^{11,12}. This may also explain the high hit rate of anti-persister adjuvants obtained from the small library used in the current study, as a significant number of chemicals, whether approved by FDA or not, are known to disrupt the bacterial membrane bilayer or proteins essential for regulating membrane potential and energy metabolism¹³. This has been discussed in the current manuscript (see lines 376-386).

2) purity and catalog numbers of the antibiotics should be reported in Methods section, the purity of the chemicals in Table S4 should also be reported

Response: The requested information is now provided in the manuscript (see Supplementary Table S4).

3) p4: comment on purchased chemical library – how do you conduct the screen if you don't know the concentrations of the drug? It seems very odd this is not supplied for a purchased library.

Response: The phenotype arrays are preconfigured 96 well plates that are extensively used by diverse research groups¹⁴⁻¹⁷. Biolog Inc. precoats the wells of the PM panels by dispensing the chemicals in appropriate solvents and then drying them in the wells (see lines 464-468). Unfortunately, the exact concentrations of these chemicals are not being disclosed by the company. Although high-throughput screening assays are known to be error prone, we use our screening strategy to identify chemical hits that are potentially lethal to persister cells. In subsequent validation experiments, we extensively analyzed these chemical hits to fully assess their utility, effectiveness, and potency by testing them for reproducibility, nearly complete inhibition of GFP expression, and eradication of persister cells (Supplementary Fig. S3 and Fig. S4).

4) p 11: full name of DiSC3(5)] incorrect – no '0' or space after thia

Response: This has been corrected.

5) p19 – no details on flow cytometry conditions provided, what x g for centrifugation?

Response: These experimental details were provided (see lines 500 and 625-640).

Figure 1: suggest splitting this into 3 Figures as the 'wall of graphs' is difficult to comprehend and each graph is too small. Separate into drug screening (A-D), phenazine characterisation (E-I) and the Pseudomonas/time dependence (J-L).

Response: This figure has been adjusted as suggested. Also, some of the panels were transferred to supplementary file (see Fig. 1, Fig. 2, Fig. 3 and Supplementary Fig. S1, Fig. S3, Fig. S4, Fig. S5, and Fig. S12).

Fig 4: again, very difficult to see. Potentially make figure full page, have timepoints run vertically rather than horizontally

Response: This figure has been adjusted as suggested (see Fig. 4 and Supplementary Fig. S6).

Reviewer #2 (Remarks to the Author):

The ms by Mohiuddin et al aims at identifying chemicals that are eradicating persister cells to ofloxacin, a fluoroquinolone inducing the SOS response. The authors screened a library of compounds first on their ability to inhibit the SOS response and then refined their criteria: reduction of the number of persister cells and low toxicity. They ended up with 7 candidates, among which 3 are phenothiazine drugs. They also tested 2 additional FDA-approved phenothiazine drugs. They follow-up their study only with phenothiazine compounds. Finding anti-persisters is certainly timely, and to my knowledge it is the first example of FDA-approved compounds. However, how in vitro persistence assays recapitulate persistence in a host is still unknown, especially for intracellular pathogens.

While the work seems promising, I found the ms difficult to follow. The rationale is not always clear, a lot of figures and supplementary figures are presented, and at the end the reader (at least I) get lost in the twists and turns of the story.

Response: We thank the reviewer for their constructive feedback. We have reduced the number of panels in the figures and explained the rationale of each experiment. Introduction, Results and Discussion have been significantly revised to make the flow better. The reviewer's other comments have been addressed point by point as described below.

Major comments:

1. I would suggest presenting the data for only one or 2 drugs (TDZ?) as main figures. This would lighten the number of panels in each figure. The others could be shown in Sup.

Response: We have split most of the figures to lighten the number of panels (see Fig. 1, Fig. 2, Fig. 3, and Fig. 4).

2. The authors should consider the time of treatment as 0h and not 5h (of culture) in all their graphs and they should mention the culture medium used. The legends are not always clear.

Response: This has been adjusted in all figures.

3. While the authors claim that they screened for drugs inhibiting the SOS response (tested with a variety of transcriptional reporters), they also show that these drugs are inhibiting translation in general. Moreover, these drugs also eradicate persister cells to a totally unrelated antibiotic, ampicillin. This indicate that these drugs are not specific to ofloxacin-induced repair mechanisms – this should be stated clearly and linked with the other effects of the drugs (membrane depolarization, ATP drop, etc). Basically, inhibition of translation appears to a side effect of the metabolic problems as transcription and/or translation inhibition on the contrary increase persistence frequency. The authors should comment on that.

Response: We agree that these drugs are not specific to ofloxacin. Although our chemical screening assay is based on the link between the SOS response and ofloxacin persistence, the identified phenothiazine drugs indirectly inhibit DNA repair mechanisms by suppressing cell metabolism (as stated by the reviewer). Our data indicate that the inhibition of transcription and translation by RIF and CM does not eliminate persister cells; however, the effects of RIF and CM on cell metabolism and physiology are not as pleiotropic as those of phenothiazine treatments. The specific points highlighted by the reviewer have been clarified several times throughout the manuscript (see lines 181-184, 186-192 and 323-331) and discussed extensively (see lines 333-348).

4. Metabolomics data show that TDZ impairs TCA cycle and ATP production and leads to accumulation of metabolites associated with glycolysis and carbon metabolism. However, in the same conditions, TDZ and other phenothiazine drugs do not impair survival. How is this possible? What are the effects on growing cells? It would be interesting to show a growth curve in the presence of TDZ.

Response: *Metabolic inhibitors (e.g., rifampicin or chloramphenicol) can be bacteriostatic. These chemicals can inhibit certain metabolic mechanisms in cells without producing any cytotoxic molecules. We would like to note that phenothiazine drugs at concentrations tested in this study become lethal when used with antibiotics; however, these drugs do not themselves reduce cell viability at the indicated concentrations (Supplementary Fig. S4). Optical density measurements of cultures indicated that the cells can continue to grow in the presence of phenothiazines, albeit at slower rates compared to untreated controls (see Fig. 3, which corresponds to supplementary Fig. S10 in the manuscript). Also, we found no detectable persister cells in E. coli MG1655 mutant strains in which the AtpA, AtpD and NuoF subunits were deleted. These are critical proteins involved in energy metabolism. Overall, our data indicate that perturbing the energy metabolism of cells (chemically or genetically) can make the cells more sensitive to antibiotics. These points were further clarified and discussed in the manuscript (see lines 131-133, 220-230, and 364-416).*

Fig. 3. Growth curves of *E. coli* MG1655 cells in the presence of a phenothiazine drug. Overnight cultures were diluted (1:1000) in fresh 25-ml LB in 250-ml flasks and cultured at 37 °C and 250 rpm. Cells were treated with thioridazine (TDZ) (0.25 mM) at indicated time points (3 h and 5 h). At designated time points samples were collected for optical density (OD₆₀₀) measurements. N=4.

5. Fig 4D: The authors show that post-treatment with TDZ do not affect persistence frequency to ofloxacin. If TDZ impairs the recovery of persisters by decreasing the metabolism, I suppose that adding TDZ after ofloxacin should also inhibit persister recovery. How do the authors explain

this observation? Does ofloxacin impair the uptake of phenothiazines? Could this be evaluated experimentally?

Response: Here, we wanted to test whether the drugs would be ineffective if the cellular repair pathways were already active. We have already shown that cells fully expressed DNA repair genes within a 1–2 h of exposure to ofloxacin in the absence of phenothiazine drugs (Fig. 1A). Therefore, we first treated cells with ofloxacin to induce the SOS response, and then added thioridazine at 1 h, 3 h, or 4 h after the addition of ofloxacin to provide sufficient time for the expression of the repair genes. The post-treatment with thioridazine did not eliminate the persister cells, which indicate that the availability of the cellular repair pathways may enhance persister survival. However, this observed phenomenon may also be due to the formation of metabolically inactive cells, as fluoroquinolones are known to induce cell dormancy through a TisB-dependent mechanism¹⁸. We have already shown that phenothiazines are not effective for metabolically inactive cells (see Fig. 7 and Supplementary Fig. S12). Although the data given in Fig. 4D is interesting and highlights the complex outcomes of post-treatment strategies (which is not our scope), we removed this data from the manuscript to improve the flow of the manuscript. As the current study has a lot of research elements, we plan to study this interesting phenomenon in depth in future.

6. Phenothiazine drugs have no effect on persistence of cells grown in stationary phase (6h of culture and more). What is the explanation? Do phenothiazines penetrate stationary phase cells? The authors should comment on that and test whether these drugs inhibit persistence of cells in log phase.

Response: We have tested these drugs on exponentially growing cells (i.e., log phase cultures), and showed that they eradicated ofloxacin persisters in these cultures (see Fig. 8c and Supplementary Fig. S12). Our research elements (e.g., untargeted metabolomics, mutant cell library screening and the subsequent validation assays that measure cellular redox activities, ATP levels and the PMF) indicate that the mechanism of action of phenothiazine drugs seems to require active metabolic machineries, and there is a very strong correlation between effectiveness of these drugs and cellular metabolism (see Fig. 7f-m and Supplementary Fig. S14). When metabolically stimulated with an exogenous carbon source (such as glucose or glycerol), intrinsically tolerant stationary-phase cells became sensitive to phenothiazines. Glucose or glycerol was previously shown to enhance electron transport chain (ETC) activities of stationary phase cells in a previous study¹⁹. In our study, *E. coli* mutant strains that were less permeabilized by phenothiazines generally lack genes encoding subunits of several ETC proteins, such as succinate: quinone oxidoreductase, cytochrome bd-I ubiquinol oxidase and NADH:quinone oxidoreductase complexes, which is, indeed, strongly supporting our argument here. These points have been clarified and discussed in related sections (see lines 254-262, 337-348 and 364-376).

7. Fig 5A: PI permeabilization induced by phenothiazines is transient. What is the explanation? Does that imply that membrane depolarization is not the main effect of these drugs leading to increase persistence? If it was the case, persistence should increase with time? What is happening in cells treated with both ofloxacin and phenothiazines?

Response: Our data shows that although phenothiazines (in the absence of ofloxacin) largely permeabilized the membranes of the bulk cell populations 1 h after the treatment, their impact was found to be transient, as the PI-positive cell levels were decreased with respect to treatment

time. This transient permeabilization does not necessarily kill the cells as the optical density measurements of cultures indicate that cells can still continue to grow in the presence of phenothiazines (but at slower rates compared to untreated controls), and eventually reach the stationary phase (see Supplementary Fig. S10). Also, when we treated the stationary phase cells with phenothiazines and ofloxacin, we were not able to kill the cells (see Supplementary Fig. S12), which is consistent with our argument above (see the response for the comment #6).

Although this transient permeabilization may increase the antibiotic uptake initially, this cannot be the only factor underlying the observed alteration in persister levels in co-treated cultures. Also, the lack of membrane permeabilization does not necessarily imply “an increase in persistence”, as we were able to identify a concentration range of thioridazine that do not permeabilize cellular membrane by themselves (Fig. 8a-b) but eliminated the persister cells when used with ofloxacin (Fig. 8a-c). These concentrations can still perturb the proton concentration gradient of the proton motive force without permeabilizing the cells. These drugs seem to inhibit a wide range of critical membrane-bound metabolic proteins, resulting in a downregulation of the proton motive force, ATP production, central carbon metabolism, and transcription and translational activities, which become greatly detrimental to antibiotic-damaged persister cells. However, these effects on persister populations in stationary phase become convoluted, as a significant number of cells from the stationary phase bulk population become intrinsically tolerant to antibiotics. Although whether these stationary phase cells are true persister cells is still under debate²⁰, the effects of phenothiazines on persistence become clearer in exponential phase cultures, as antibiotics sterilize the bulk cell population. These points have been clarified and discussed in the manuscript (see lines 286-298, and 334-348).

8. The rationale behind the experiments presented in Fig 5B is unclear (PI permeability of KO mutants).

Response: Our rationale is if there is an interaction between a membrane protein and thioridazine, which enhances cellular membrane permeability, membrane permeability should be significantly reduced in the absence of this membrane protein (see lines 236-239).

9. Fig 6C: It seems that some bar duplications have occurred. The blue bars represent the TDZ treatment at different concentrations. As far as I can see, bars at 0.1mM, 0.15mM and 0.2mM are identical and those at 0mM and 0.25mM as well. It seems to me that the data should be presented in one single graph as untreated and ofloxacin controls appear to be the same for the different conditions.

Response: The figures were updated as suggested (see Fig. 8c).

10. Figure 6D, E, F and related text are not clear. The text seems to be rather a discussion than what is needed to introduce the rationale of the presented experiments. Fig 6D appears to be already published? Is that correct? Did the authors redo the experiment in the conditions used in this ms?

Response: The data in Fig. 6D has been generated in this study although the concept describing this figure has been already published and discussed in our previous papers^{9,21}. To prevent any possible confusion, we removed this figure from our manuscript. Also, the text mentioned by the reviewer was clarified and moved to the discussion section now (see lines 364-378).

11. L366: Not sure why the authors postulate that depletion of ATP inhibits repair. Their data show that translation is inhibited by phenothiazines, implying that repair proteins are not

synthesized. This is observed in Fig4.

Response: This sentence was removed.

12. At the end, what are the main conclusions about the mechanisms of action of those drugs? Why are the effects only revealed in the presence of ofloxacin? What is happening in the presence of ampicillin? Why the effect of phenothiazines is different in cells grown for 5h (early stationary phase) and grown longer? Does that imply that persisters arising from growing cells and stationary phase cells are different? Does it suggest that the persistence mechanisms are different? Are stationary phase ‘true’ persisters or ‘just’ stationary phase cells that are refractory to a lot of different stress – antibiotics, phenothiazines? By which mechanisms could the synergy between ofloxacin and phenothiazines be explained? What is(are) the mechanisms behind the need to add both compounds at the same time to observe an effect of phenothiazines on persistence frequency? The discussion section should address all these points and put the observations in perspective with the current knowledge regarding persistence. I also suggest to improve the introduction by providing a more general view of the mechanisms involved in persistence and not to focus only on the SOS response as it seems very unlikely that phenothiazines specifically target DNA repair.

Response: We have extensively edited our manuscript to discuss all the points raised by the reviewer (see the entire Discussion section). We also significantly update the introduction, which has more general view of the mechanisms involved in persistence (see lines 29-52). We also eliminated some texts from the manuscript that were excessively focusing on the SOS response. Both manuscript files, with or without tracked changes, were provided to verify that significant changes haven been made.

Minor comments:

Even though ‘resuscitation’ is used by some groups in the persistence field, I think it is not appropriate in a scientific context.

Response: In the current manuscript, we used the “persister recovery” term instead of “persister resuscitation”, consistent with a previously published study²².

Maybe original papers instead of too many reviews should be cited in the introduction

Response: References significantly updated.

L44: I don’t understand what the authors mean by ‘persisters are capable of physiological responses to external factors that are essential for resuscitation after antibiotic removal’. Should be more explicit.

Response: The related section has been corrected as follows: “persisters are capable of responding to external factors, indicating the presence of active cellular processes that may be essential for their recovery” (see lines 60-61).

References

1. Mohiuddin, S. G., Hoang, T., Saba, A., Karki, P. & Orman, M. A. Identifying Metabolic Inhibitors to Reduce Bacterial Persistence. *Front. Microbiol.* **11**, 472 (2020).
2. Balaban, N. Q., Merrin, J., Chait, R., Kowalik, L. & Leibler, S. Bacterial persistence as a

- phenotypic switch. *Science*. **305**, 1622–1625 (2004).
3. Weinstein, E. A. *et al.* Inhibitors of type II NADH:menaquinone oxidoreductase represent a class of antitubercular drugs. *Proc. Natl. Acad. Sci. U. S. A.* **102**, 4548–4553 (2005).
 4. Yano, T., Lin-Sheng, L., Weinstein, E., Teh, J. S. & Rubin, H. Steady-state kinetics and inhibitory action of antitubercular phenothiazines on *Mycobacterium tuberculosis* Type-II NADH-menaquinone oxidoreductase (NDH-2). *J. Biol. Chem.* **281**, 11456–11463 (2006).
 5. Nzakizwanayo, J. *et al.* Fluoxetine and thioridazine inhibit efflux and attenuate crystalline biofilm formation by *Proteus mirabilis*. *Sci. Rep.* **7**, 1–14 (2017).
 6. De Keijzer, J. *et al.* Thioridazine alters the cell-envelope permeability of *Mycobacterium tuberculosis*. *J. Proteome Res.* **15**, 1776–1786 (2016).
 7. Kaatz, G. W., Moudgal, V. V., Seo, S. M. & Kristiansen, J. E. Phenothiazines and thioxanthenes inhibit multidrug efflux pump activity in *Staphylococcus aureus*. *Antimicrob. Agents Chemother.* **47**, 719–726 (2003).
 8. Stokes, J. M., Lopatkin, A. J., Lobritz, M. A. & Collins, J. J. Bacterial Metabolism and Antibiotic Efficacy. *Cell Metab.* **30**, 251–259 (2019).
 9. Orman, M. A. & Brynildsen, M. P. Inhibition of stationary phase respiration impairs persister formation in *E. coli*. *Nat. Commun.* **6**, 1–13 (2015).
 10. Orman, M. A. & Brynildsen, M. P. Persister formation in *Escherichia coli* can be inhibited by treatment with nitric oxide. *Free Radic. Biol. Med.* **93**, 145–154 (2016).
 11. Stokes, J. M. *et al.* A Deep Learning Approach to Antibiotic Discovery. *Cell* **180**, 688–702.e13 (2020).
 12. Wang, M., Chan, E. W. C., Wan, Y., Wong, M. H. yin & Chen, S. Active maintenance of proton motive force mediates starvation-induced bacterial antibiotic tolerance in *Escherichia coli*. *Commun. Biol.* **4**, 1–11 (2021).
 13. Hurdle, J. G., O’Neill, A. J., Chopra, I. & Lee, R. E. Targeting bacterial membrane function: an underexploited mechanism for treating persistent infections. *Nat. Rev. Microbiol.* **9**, 62–75 (2010).
 14. Blanco, P., Corona, F. & Martínez, J. L. Biolog phenotype microarray is a tool for the identification of multidrug resistance efflux pump inducers. *Antimicrob. Agents Chemother.* **62**, (2018).
 15. Jałowicki, Ł. *et al.* Using phenotype microarrays in the assessment of the antibiotic susceptibility profile of bacteria isolated from wastewater in on-site treatment facilities. *Folia Microbiol. (Praha)*. **62**, 453–461 (2017).
 16. Palmer, J. M., Drees, K. P., Foster, J. T. & Lindner, D. L. Extreme sensitivity to ultraviolet light in the fungal pathogen causing white-nose syndrome of bats. *Nat. Commun.* **9**, 1–10 (2018).
 17. Fondi, M. *et al.* Genomic and phenotypic characterization of the species *Acinetobacter venetianus*. *Sci. Rep.* **6**, 1–12 (2016).
 18. Dörr, T., Vulić, M. & Lewis, K. Ciprofloxacin Causes Persister Formation by Inducing the TisB toxin in *Escherichia coli*. *PLoS Biol.* **8**, e1000317 (2010).

19. Allison, K. R., Brynildsen, M. P. & Collins, J. J. Metabolite-enabled eradication of bacterial persisters by aminoglycosides. *Nature*. **473**, 216–220 (2011).
20. Balaban, N. Q. *et al.* Definitions and guidelines for research on antibiotic persistence. *Nat. Rev. Microbiol.* **17**, 441–448 (2019).
21. Orman, M. A. & Brynildsen, M. P. Dormancy is not necessary or sufficient for bacterial persistence. *Antimicrob. Agents Chemother.* **57**, 3230–3239 (2013).
22. Völzing, K. G. & Brynildsen, M. P. Stationary-phase persisters to ofloxacin sustain DNA damage and require repair systems only during recovery. *MBio.* **6**, e00731-15 (2015).

REVIEWERS' COMMENTS:

Reviewer #1 (Remarks to the Author):

The authors have substantively addressed the reviewer comments, and the manuscript is significantly improved. While I still feel strongly that in vivo confirmation is really needed to provide high impact, by demonstrating that this in vitro observation can translate into useful treatment potential, the additional in vitro experiments the authors have provided does increase the significance.

One minor comment is that the flow cytometry experimental conditions are still not sufficient - the type of flow cytometer and flow rates are not provided, and for In 576 and In 615, the number of events recorded.

Reviewer #3 (Remarks to the Author):

The revised version of the manuscript appears to be much improved; in particular, there is now a clearer presentation of the biological context of their findings, and there is a more honest discussion of caveats. While the exact mechanism of action of their potentiators remains unknown, this is none the less a strong contribution.

As a minor last comment, I'd prefer if they did not call their phenotype persister "eradication". I know this term has become quite popular in the field, but with their current limit of detection, it is not quite accurate (I would accept "eradication" if they had plated 10 mL of treated cultures and found no survivors). Please replace with "strong reduction" or something similar.

Reviewer #1 (Remarks to the Author):

The authors have substantively addressed the reviewer comments, and the manuscript is significantly improved. While I still feel strongly that in vivo confirmation is really needed to provide high impact, by demonstrating that this in vitro observation can translate into useful treatment potential, the additional in vitro experiments the authors have provided does increase the significance.

One minor comment is that the flow cytometry experimental conditions are still not sufficient - the type of flow cytometer and flow rates are not provided, and for Ln 576 and Ln 615, the number of events recorded.

Response: We would like to thank the reviewer for their positive feedback. We have provided additional details about our flow cytometry analysis, as requested (see the “PI Staining” section in Methods, and Supplementary Fig. 8).

Reviewer #3 (Remarks to the Author):

The revised version of the manuscript appears to be much improved; in particular, there is now a clearer presentation of the biological context of their findings, and there is a more honest discussion of caveats. While the exact mechanism of action of their potentiators remains unknown, this is none the less a strong contribution.

As a minor last comment, I'd prefer if they did not call their phenotype persister "eradication". I know this term has become quite popular in the field, but with their current limit of detection, it is not quite accurate (I would accept "eradication" if they had plated 10 mL of treated cultures and found no survivors). Please replace with "strong reduction" or something similar.

Response: We would like to thank the reviewer for their effort in reviewing our manuscript. We removed the term “eradication” throughout the manuscript and used “reduction”.